# Canonical neural networks perform active inference

Takuya Isomura [1✉], Hideaki Shimazaki [2] & Karl J. Friston [3]

This work considers a class of canonical neural networks comprising rate coding models, wherein neural activity and plasticity minimise a common cost function—and plasticity is modulated with a certain delay. We show that such neural networks implicitly perform active inference and learning to minimise the risk associated with future outcomes. Mathematical analyses demonstrate that this biological optimisation can be cast as maximisation of model evidence, or equivalently minimisation of variational free energy, under the well-known form of a partially observed Markov decision process model. This equivalence indicates that the delayed modulation of Hebbian plasticity—accompanied with adaptation of firing thresholds—is a sufficient neuronal substrate to attain Bayes optimal inference and control. We corroborated this proposition using numerical analyses of maze tasks. This theory offers a universal characterisation of canonical neural networks in terms of Bayesian belief updating and provides insight into the neuronal mechanisms underlying planning and adaptive behavioural control.

[1] Brain Intelligence Theory Unit, RIKEN Center for Brain Science, Wako, Saitama 351-0198, Japan. [2] Center for Human Nature, Artificial Intelligence, and Neuroscience (CHAIN), Hokkaido University, Sapporo, Hokkaido 060-0812, Japan. [3] Wellcome Centre for Human Neuroimaging, Institute of Neurology, University College London, 12 Queen Square, London, WC1N 3AR, UK. ✉email: takuya.isomura@riken.jp

The sentient behaviour of biological organisms is characterised by optimisation. Biological organisms recognise the state of their environment by optimising internal representations of the external (i.e. environmental) dynamics generating sensory inputs. In addition, they optimise their behaviour for adaptation to the environment, thereby increasing their probability of survival and reproduction. This biological self-organisation is typically formulated as the minimisation of cost functions[1–3], wherein a gradient descent on a cost function furnishes neural dynamics and synaptic plasticity. However, two fundamental issues remain to be established—namely, the characterisation of the dynamics of an arbitrary neural network as a generic optimisation process—and the correspondence between such neural dynamics and statistical inference[4] found in applied mathematics and machine learning. The present work addresses these issues by demonstrating that a class of canonical neural networks of rate coding models is functioning as—and thus universally characterised in terms of—variational Bayesian inference, under a particular but generic form of the generative model.

Variational Bayesian inference offers a unified explanation for inference, learning, prediction, decision making, and the evolution of biological form[5,6]. This kind of inference rests upon a generative model that expresses a hypothesis about the generation of sensory inputs. Perception and behaviour can then be read as optimising the evidence for a 'generative model', inherent in sensory exchanges with the environment. The ensuing evidence lower bound (ELBO)[7]—or equivalently variational free energy, which is the negative of the ELBO—then plays the role of a cost function. Variational free energy is the standard cost function in variational Bayes—and provides an upper bound on surprise (i.e., improbability) of sensory inputs. Minimisation of variational free energy, with respect to internal representations, then yields approximate posterior beliefs about external states. Similarly, the minimisation of variational free energy with respect to action on external states maximises the evidence or marginal likelihood of resulting sensory samples. This framework integrates perceptual (unsupervised), reward-based (reinforcement), and motor (supervised) learning in a unified formulation that shares many commitments with neuronal implementations of approximate Bayesian inference using spiking models[8–12]. In short, internal states of an autonomous system under a (possibly nonequilibrium) steady state can be viewed as parameterising posterior beliefs of external states[13–15]. In particular, active inference aims to optimise behaviours of a biological organism to minimise a certain kind of risk in the future[16–18], wherein risk is typically expressed in a form of expected free energy (i.e., the variational free energy expected under posterior predictive beliefs about the outcomes of a given course of action).

Crucially, as a corollary of the complete class theorem[19–21], any neural network minimising a cost function can be viewed as performing variational Bayesian inference, under some prior beliefs. We have previously introduced a reverse-engineering approach that identifies a class of biologically plausible cost functions for neural networks[22]. This foundational work identified a class of cost functions for single-layer feedforward neural networks of rate coding models with a sigmoid (or logistic) activation function—based on the assumption that the dynamics of neurons and synapses follow a gradient descent on a common cost function. We subsequently demonstrated the mathematical equivalence between the class of cost functions for such neural networks and variational free energy under a particular form of the generative model. This equivalence licences variational Bayesian inference as a fundamental optimisation process that underlies both the dynamics and function of such neural networks. Moreover, it enables one to characterise any variables and

constants in the network in terms of quantities (e.g. priors) that underwrite variational Bayesian inference[22]. However, it remains to be established whether the active inference is an apt explanation for any given neural network that actively exchanges with its environment. In this paper, we address this enactive or control aspect to complete the formal equivalence of neural network optimisation and the free-energy principle.

In most formulations, active inference goes further than simply assuming action and perception minimise variational free energy—it also considers the consequences of action as minimising expected free energy, i.e. planning (and control) as inference, as a foundational approach to sentient behaviour[12,23–27]. Thus, to evince active inference in neural networks, it is necessary to demonstrate that they can plan to minimise future risks.

To address this issue, this work identifies a class of biologically plausible cost functions for two-layer recurrent neural networks, under an assumption that neural activity and plasticity minimise a common cost function (referred to as assumption 1). Then, we analytically and numerically demonstrate the implicit ability of neural networks to plan and minimise future risk, when viewed through the lens of active inference. Namely, we suppose a network of rate coding neurons with a sigmoid activation function, wherein the middle layer involves recurrent connections, and the output layer provides feedback responses to the environment (assumption 2). In this work, we will call such architectures canonical neural networks (Table 1). Then, we demonstrate that the class of cost functions—describing their dynamics—can be cast as variational free energy under an implicit generative model, in the well-known form of a partially observable Markov decision process (POMDP). The gradient descent on the ensuing cost function naturally yields Hebbian plasticity[28–30] with an activity-dependent homoeostatic term.

In particular, we consider the case where an arbitrary modulator[31–33] regulates synaptic plasticity with a certain delay (assumption 3) and demonstrate that such modulation is identical to the update of a policy through a post hoc evaluation of past decisions. The modulator renders the implicit cost function a risk function, which in turn renders behavioural control Bayes optimal—to minimise future risk. The proposed analysis affirms that active inference is an inherent property of canonical neural networks exhibiting delayed modulation of Hebbian plasticity. We discuss possible neuronal substrates that realise this modulation.

## Results

**Overview of equivalence between neural networks and variational Bayes.** First, we summarise the formal correspondence between neural networks and variational Bayes. A biological agent is formulated here as an autonomous system comprising a network of rate coding neurons (Fig. 1a). We presume that neural activity, action (decision), synaptic plasticity, and changes in any other free parameters minimise a common cost function $L := L(o_{1:t}, \varphi)$ (c.f., assumption 1 specified in Introduction). Here, $o_{1:t} := \{o_1, \ldots, o_t\}$ is a sequence of observations and $\varphi := \{x_{1:t}, y_{1:t}, W, \phi\}$ is a set of internal states comprising the middle-layer ($x_\tau$) and output-layer ($y_\tau$) neural activity, synaptic strengths ($W$), and other free parameters ($\phi$) that characterise $L$ (e.g. firing threshold). Output-layer activity $y_t$ determines the network's actions or decisions $\delta_t$. Based on assumption 1 and the continuous updating nature of $\varphi$, the update rule for the $i$-th component of $\varphi$ is derived as the gradient descent on the cost function, $\dot{\varphi}_i \propto -\partial L / \partial \varphi_i$. This determines the dynamics of neural networks, including their activity and plasticity.

In contrast, variational Bayesian inference depicts a process of updating the prior distribution of external states $P(\vartheta)$ to the corresponding posterior distribution $Q(\vartheta)$ based on a sequence of observations. Here, $Q(\vartheta)$ approximates (or possibly exactly equals

**Table 1 Glossary of expressions.**

| Expression | Description |
|---|---|
| Canonical neural network | In this work, a canonical neural network is defined by differential equations of neural activity derived as a reduction of realistic neuron models through some approximations, which give a network of rate coding neurons with a sigmoid activation function. In particular, we consider networks comprising a middle layer that involves recurrent connections and the output layer that provides feedback responses to the environment. |
| $o_t, s_t, \delta_t$ | Observations $o_t$, hidden states $s_t$ and decisions (actions) $\delta_t$ are random variables that follow categorical distributions. Each element of them takes 0 or 1. |
| $\Gamma_t$ | Risk function $\Gamma_t$ parameterises a categorical distribution over $\gamma_t$: $P(\gamma_t) = \mathrm{Cat}((\Gamma_t, \overline{\Gamma_t})^\top)$. This can be read as an arbitrary neuromodulator that regulates synaptic plasticity through Eq. (9), which becomes a risk function under the variational Bayes formulation. |
| $A, B, C$ | Parameter matrices $A$, $B$, and $C$ are random variables with Dirichlet distributions $P(A) = \mathrm{Dir}(a)$, $P(B) = \mathrm{Dir}(b)$ and $P(C) = \mathrm{Dir}(c)$. |
| $\mathrm{Cat}(A)$ | Categorical distribution. In this expression, the probability that $o_\tau = i$ occurs given $s_\tau = j$ is $A_{ij}$; that is, $P(o_\tau = i \mid s_\tau = j, A) = A_{ij}$. For any pair of $i$ and $j$, this is expressed as $P(o_\tau \mid s_\tau, A) = \mathrm{Cat}(A)$, where matrix $A$ is the likelihood mapping that maps $s_\tau$ to $o_\tau$. Although one may prefer to denote it as $\mathrm{Cat}(As_\tau)$ to emphasise that it changes depending on $s_\tau$, in this paper, we use $\mathrm{Cat}(A)$ following the notation in previous work. Note that $A_{kl}^{(i)}$ means that the probability that $o_\tau^{(i)} = k$ occurs given $s_\tau = \vec{l} = (l_1, \ldots, l_N)^\top \in \{0,1\}^{N_s}$. |

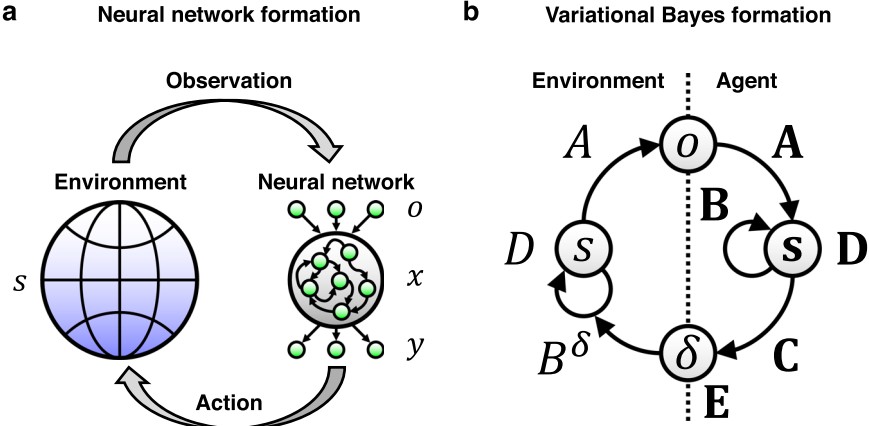

**a   Neural network formation**

Observation

Environment   Neural network

$s$   $o$   $x$   $y$

Action

**b   Variational Bayes formation**

Environment   Agent

$A$   $o$   $A$

$D$   $s$   $B$   $s$   $D$

$B$   $\delta$   $C$

$\delta$   $E$

**Fig. 1 Schematic of an external milieu and neural network, and the corresponding Bayesian formation. a** Interaction between the external milieu and autonomous system comprising a two-layer neural network. On receiving sensory inputs or observations $o(t)$ that are generated from hidden states $s(t)$, the network activity $x(t)$ generates outputs $y(t)$. The gradient descent on a neural network cost function $L$ determines the dynamics of neural activity and plasticity. Thus, $L$ is sufficient to characterise the neural network. The proposed theory affirms that the ensuing neural dynamics are self-organised to encode the posterior beliefs about hidden states and decisions. **b** Corresponding variational Bayesian formation. The interaction depicted in **a** is formulated in terms of a POMDP model, which is parameterised by $A, B, C \in \theta$ and $D, E \in \lambda$. Variational free energy minimisation allows an agent to self-organise to encode the hidden states of the external milieu—and to make decisions minimising future risk. Here, variational free energy $F$ is sufficient to characterise the inferences and behaviours of the agent.

to) $P(\vartheta \mid o_{1:t})$. This process is formulated as a minimisation of the surprise of past-to-present observations—or equivalently maximisation of the model evidence—which is attained by minimising variational free energy as a tractable proxy. We suppose that the generative model $P(o_{1:t}, \vartheta)$ is characterised by a set of external states, $\vartheta := \{s_{1:t}, \delta_{1:t}, \theta, \lambda\}$, comprising hidden states ($s_\tau$), decision ($\delta_\tau$), model parameters ($\theta$) and hyper parameters ($\lambda$) (Fig. 1b). Based on the given generative model, variational free energy is defined as a function of $Q(\vartheta)$ as follows: $F(o_{1:t}, Q(\vartheta)) := E_{Q(\vartheta)}[-\ln P(o_{1:t}, \vartheta) + \ln Q(\vartheta)]$. Here, $E_{Q(\vartheta)}[\cdot] := \int \cdot\, Q(\vartheta) d\vartheta$ denotes the expectation over $Q(\vartheta)$. In particular, we assume that $Q(\vartheta)$ is an exponential family (as considered in previous works[10]) and the posterior expectation of $\vartheta$, $\pmb{\vartheta} := E_{Q(\vartheta)}[\vartheta]$, or its counterpart, are the sufficient statistics that parameterise (i.e. uniquely determine) $Q(\vartheta)$. Under this condition, $F$ is reduced to a function of $\pmb{\vartheta}$, $F = F(o_{1:t}, \pmb{\vartheta})$. The variational update rule for the $i$-th component of $\pmb{\vartheta}$ is given as the gradient descent on variational free energy, $\dot{\pmb{\vartheta}}_i \propto -\partial F / \partial \pmb{\vartheta}_i$.

Crucially, according to the complete class theorem, a dynamical system that minimises its cost function can be viewed as performing Bayesian inference under some generative model and prior beliefs. The complete class theorem[19–21] states that for any pair of admissible decision rules and cost functions, there is some generative model with prior beliefs that renders the decisions Bayes optimal (refer to Supplementary Methods 1 for technical details). Thus, this theorem ensures the presence of a generative model that formally corresponds to the above-defined neural network characterised by $L$. Hence, this speaks to the equivalence between the class of neural network cost functions and variational free energy under such a generative model:

$$L(o_{1:t}, \varphi) \equiv F(o_{1:t}, \pmb{\vartheta}) \qquad (1)$$

wherein the internal states of a network $\varphi$ encode or parameterise the posterior expectation $\pmb{\vartheta}$. This mathematical equivalence means that an arbitrary neural network, in the class under consideration, is implicitly performing active inference through variational free

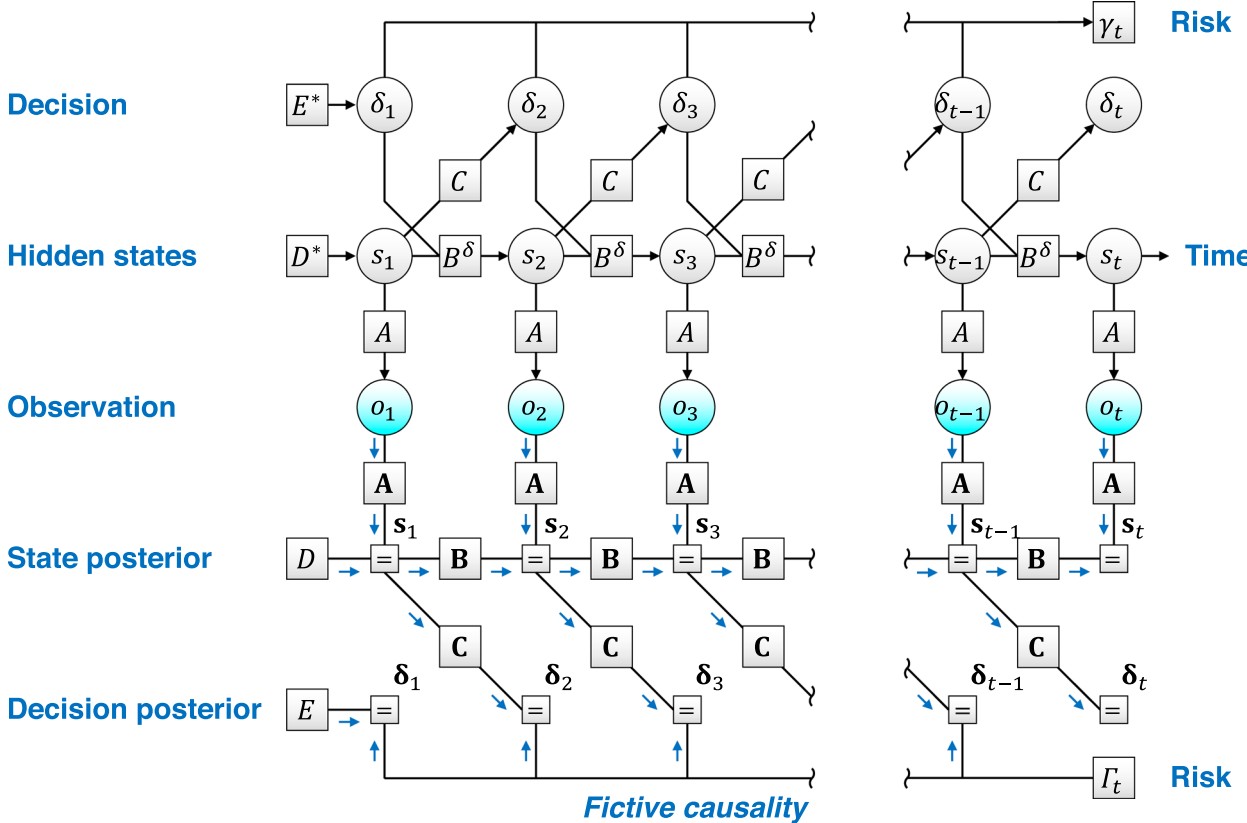

**Fig. 2 Factor graph depicting a fictive causality of factors that the generative model hypothesises.** The POMDP model is expressed as a Forney factor graph[69,70] based upon the formulation in ref. [71]. The arrows from the present risk $\gamma_t$—sampled from $\Gamma_t$—to past decisions $\delta_\tau$ optimise the policy in a post hoc manner, to minimise future risk. In reality, the current error $\gamma_t$ is determined based on past decisions (top). In contrast, decision making to minimise the future risk implies a fictive causality from $\gamma_t$ to $\delta_\tau$ (bottom). Inference and learning correspond to the inversion of this generative model. Postdiction of past decisions is formulated as the learning of the policy mapping, conditioned by $\gamma_t$. Here, A, B and C indicate matrices of the conditional probability, and bold case variables are the corresponding posterior beliefs. Moreover, $D^*$ and $E^*$ indicate the true prior beliefs about hidden states and decisions, while $D$ and $E$ indicate the priors that the network operates under. When and only when $D = D^*$ and $E = E^*$, inferences and behaviours are optimal for a given task or set of environmental contingencies, and are biased otherwise.

energy minimisation. Minimisation of $F$ is achieved when and only when the posterior beliefs best match the true conditional probability of the external states. Thus, the dynamics that minimise $L$ must induce a recapitulation of the external states in the internal states of the neural network. This is a fundamental aspect of optimisation in neural networks. This notion is essential to understand the functional meaning of the dynamics evinced by an arbitrary neural network, which is otherwise unclear by simply observing the network dynamics.

Note that being able to characterise the neural network in terms of maximising model evidence lends it an 'explainability', in the sense that the internal (neural network) states and parameters encode Bayesian beliefs or expectations about the causes of observations. In other words, the generative model explains how outcomes were generated. However, the complete class theorem does not specify the form of a generative model for any given neural network. To address this issue, in the remainder of the paper, we formulate active inference using a particular form of POMDP models, whose states take binary values. This facilitates the identification of a class of generative models that corresponds to a class of canonical neural networks—comprising rate coding models with the sigmoid activation function.

**Active inference formulated using a postdiction of past decisions.** In this section, we define a generative model and ensuing

variational free energy that corresponds to a class of canonical neural networks that will be considered in the subsequent section. The external milieu is expressed as a discrete state space in the form of a POMDP (Fig. 2). The generation of observations $o_\tau := (o_\tau^{(1)}, \dots, o_\tau^{(N_o)})^{\mathrm{T}}$ from external or hidden states milieu $s_\tau := (s_\tau^{(1)}, \dots, s_\tau^{(N_s)})^{\mathrm{T}}$ is expressed in the form of a categorical distribution, $P(o_\tau|s_\tau, A) = \mathrm{Cat}(A)$, where matrix $A$ is also known as the likelihood mapping (Table 1). Here, $A_{ij}$ means the probability that $o_\tau = i$ is realised given $s_\tau = j$, and $1 \le \tau \le t$ denotes an arbitrary time in the past or the present. Our agent receives $o_\tau$, infers latent variables (hidden states) $s_\tau$, and provides a feedback decision $\delta_\tau := (\delta_\tau^{(1)}, \dots, \delta_\tau^{(N_\delta)})^{\mathrm{T}}$ to the external milieu. Thus, the state transition at time $\tau$ depends on the previous decision $\delta_{\tau-1}$, characterised by the state transition matrix $B^\delta$, $P(s_\tau|s_{\tau-1}, \delta_{\tau-1}, B) = \mathrm{Cat}(B^\delta)$. Moreover, decision at time $\tau$ is conditioned on the previous state $s_{\tau-1}$ and current risk $\gamma_t$, characterised by the policy mapping $C$, $P(\delta_\tau|s_{\tau-1}, \gamma_t, C)$, where $\gamma_t$ contextualises $P(\delta_\tau|s_{\tau-1}, \gamma_t, C)$ as described below. Each element of $s_\tau$, $o_\tau$, and $\delta_\tau$ adopts a binary value, which is suitable for characterising generative models implicit in canonical neural networks. Note that when dealing with external states that factorise (e.g. what and where), block matrices $A$, $B$ and $C$ are the outer products of submatrices (please refer to Methods section 'Generative model' for further details; see

also ref. [22]). Hence, we define the generative model as follows:

$$P(o_{1:t}, \delta_{1:t}, s_{1:t}, \gamma_t, \theta)$$
$$= P(\theta)P(\gamma_t)\prod_{\tau=1}^{t} P(o_\tau|s_\tau, A)P(s_\tau|s_{\tau-1}, \delta_{\tau-1}, B)P(\delta_\tau|s_{\tau-1}, \gamma_t, C) \quad (2)$$

where $\theta := \{A, B, C\}$ constitute the set of parameters, and $P(s_1|s_0, \delta_0, B) = P(s_1)$ and $P(\delta_1|s_0, \gamma_t, C) = P(\delta_1)$ denote probabilities at $\tau = 1$. $P(\theta)$ and $P(\gamma_t)$ are the prior distributions of the parameters and the risk, which implies that $\theta$ and $\gamma_t$ are treated as random variables in this work (Table 1). Initial states and decisions are characterised by prior distributions $P(s_1) = \text{Cat}(D)$ and $P(\delta_1) = \text{Cat}(E)$, where $D$ and $E$ are block vectors.

The agent makes decisions to minimise a risk function $\Gamma_t := \Gamma(o_{1:t}, \mathbf{s}_{1:t}, \boldsymbol{\delta}_{1:t-1}, \theta)$ that it employs (where $0 \le \Gamma_t \le 1$; see Table 1). Because the current risk $\Gamma_t$ is a consequence of past decisions, the agent needs to select decisions that minimise the future risk. In this sense, $\Gamma_t$ is associated with the expected free energy and precision in the usual formulation of active inference[17,18] (see Methods section 'Generative model' and Supplementary Methods 2 for details).

To characterise optimal decisions as minimising expected risk, in our POMDP model, we use a fictive mapping from the current risk $\Gamma_t$ to past decisions $\delta_1, \dots, \delta_{t-1}$ (Fig. 2). Although this is not the true causality in the real generative process that generates sensory data, here we intend to model the manner that an agent subjectively evaluates its previous decisions after experiencing their consequences. This fictive causality is expressed in the form of a categorical distribution,

$$P(\delta_\tau|s_{\tau-1}, \gamma_t, C) = \text{Cat}(C)^{\overline{\gamma_t}}\text{Cat}(C' \oslash C)^{\gamma_t} \quad (3)$$

wherein policy mapping $C$ is switched by a binarized risk $\gamma_t \in \{0, 1\}$—sampled from $P(\gamma_t) = \text{Cat}((\Gamma_t, \overline{\Gamma_t})^{\text{T}})$—in a form of mixture model. We select this form of generative model because it speaks to the neuromodulation of synaptic plasticity, as shown in the next section. Equation (3) says that the probability of selecting a decision $\delta_\tau$ after receiving $s_{\tau-1}$ is determined by matrix $C$ when $\gamma_t = 0$, whereas it is inversely proportional to $C$ (in the element-wise sense) when $\gamma_t = 1$. We note that matrix $C'$ denotes a normalisation factor that can be dropped from the following formulations, and $\oslash$ indicates the element-wise division operator. Throughout the manuscript, the overline variable indicates one minus the variable; e.g. $\overline{\gamma_t} = 1 - \gamma_t$.

Importantly, the agent needs to keep selecting 'good' decisions while avoiding 'bad' decisions. To this end, Eq. (3) supposes that the agent learns from the failure of decisions, by assuming that the bad decisions were sampled from the opposite of the optimal policy mapping. In other words, the agent is assumed to have the prior belief such that the decision—sampled from $\text{Cat}(C)$—should result in $\gamma_t = 0$, while sampling from $\text{Cat}(C' \oslash C)$ should yield $\gamma_t = 1$. This construction enables the agent to conduct a postdiction of its past decisions—and thereby to update the policy mapping to minimise future risk—by associating the past decision rule (policy) with the current risk. Further details are provided in the Methods section 'Generative model'. In the next section, we will explain the biological plausibility of this form of adaptive behavioural control, wherein the update of the policy mapping turns out to be identical to a delayed modulation of Hebbian plasticity.

In short, Eq. (3) presumes that a past decision $\delta_\tau$ ($1 \le \tau \le t - 1$) is determined based on a past state $s_{\tau-1}$ and the current risk $\gamma_t$. In contrast, the current decision $\delta_t$ is determined to minimise the future risk, $P(\delta_t|s_{t-1}, C) = \text{Cat}(C)$, because the agent has not yet observed the consequences of the current decision. We note that

although by convention, active inference uses $C$ to denote the prior preference, this work uses $C$ to denote a mapping to determine a decision depending on the previous state. Herein, the prior preference is implicit in the risk function $\Gamma_t$. Due to construction, $C'$ does not explicitly appear in the inference; thus, it is omitted in the following formulations.

Variational Bayesian inference refers to the process that optimises the posterior belief $Q(\vartheta)$. Based on the mean-field approximation, $Q(\vartheta)$ is expressed as

$$Q(\vartheta) = Q(A)Q(B)Q(C)\prod_{\tau=1}^{t} Q(s_\tau)Q(\delta_\tau) \quad (4)$$

Here, the posterior beliefs about states and decisions are categorical probability distributions, $Q(s_\tau) = \text{Cat}(\mathbf{s}_\tau)$ and $Q(\delta_\tau) = \text{Cat}(\boldsymbol{\delta}_\tau)$, whereas those about parameters are Dirichlet distributions, $Q(A) = \text{Dir}(\mathbf{a})$, $Q(B) = \text{Dir}(\mathbf{b})$, and $Q(C) = \text{Dir}(\mathbf{c})$. Throughout the manuscript, bold case variables (e.g. $\mathbf{s}_\tau$) denote the posterior expectations of the corresponding italic case random variables (e.g. $s_\tau$). Thus, $\mathbf{s}_\tau$ forms a block vector that represents the posterior probabilities of elements of $s_\tau$ taking 1 or 0. The agent samples a decision $\delta_t$ at time $t$ from the posterior distribution $Q(\delta_t)$. In this paper, the posterior belief of transition mapping is averaged over all possible decisions, $\mathbf{B} = \text{E}_{Q(\delta)}[\mathbf{B}^\delta]$, to ensure the exact correspondence to canonical neural networks. We use $\theta := \{\mathbf{a}, \mathbf{b}, \mathbf{c}\}$ to denote the parameter posteriors. For simplicity, here we suppose that state and decision priors ($D, E$) are fixed.

Under the above-defined generative model and posterior beliefs, the ensuing variational free energy is analytically expressed as follows:

$$F(o_{1:t}, \mathbf{s}_{1:t}, \boldsymbol{\delta}_{1:t}, \theta) = \sum_{\tau=1}^{t} \mathbf{s}_\tau \cdot (\ln \mathbf{s}_\tau - \ln \mathbf{A} \cdot o_\tau - \ln \mathbf{B}\mathbf{s}_{\tau-1})$$
$$+ \sum_{\tau=1}^{t} \boldsymbol{\delta}_\tau \cdot (\ln \boldsymbol{\delta}_\tau - (1 - 2\Gamma_{t,\tau})\ln \mathbf{C}\mathbf{s}_{\tau-1}) + \mathcal{O}(\ln t) \quad (5)$$

The derivation details are provided in the Methods section 'Variational free energy'. Note that $\Gamma_{t,\tau} = 0$ for $\tau = t$; otherwise, $\Gamma_{t,\tau} = \Gamma_t$. The order $\ln t$ term indicates the complexity of parameters, which is negligible when the leading order term is large. The gradient descent on variational free energy updates the posterior beliefs about hidden states ($\mathbf{s}_t$), decisions ($\boldsymbol{\delta}_t$) and parameters ($\theta$). The optimal posterior beliefs that minimise variational free energy are obtained as the fixed point of the implicit gradient descent, which ensures that $\partial F/\partial \mathbf{s}_t = 0$, $\partial F/\partial \boldsymbol{\delta}_t = 0$ and $\partial F/\partial \theta = O$. The explicit forms of the posterior beliefs are provided in the Methods section 'Inference and learning'.

To explicitly demonstrate the formal correspondence with the cost functions for neural networks considered in the next section, we further transform the variational free energy as follows: based on Bayes theorem $P(s_\tau|s_{\tau-1}, B^\delta) \propto P(s_{\tau-1}|s_\tau, B^\delta)P(s_\tau)$, the inverse transition mapping is expressed as $\mathbf{B}^\dagger = \mathbf{B}^{\text{T}}\text{diag}[D]^{-1}$ using the state prior $P(s_\tau) = \text{Cat}(D)$ (where $P(s_{\tau-1})$ is supposed to be a flat prior belief). Moreover, from Bayes theorem $P(\delta_\tau|s_{\tau-1}, \gamma_t, C) \propto P(s_{\tau-1}|\delta_\tau, \gamma_t, C)P(\delta_\tau)$, the inverse policy mapping is expressed as $\mathbf{C}^\dagger = \mathbf{C}^{\text{T}}\text{diag}[E]^{-1}$ using the decision prior $P(\delta_t) = \text{Cat}(E)$. Using these relationships, Eq. (5) is transformed into the form shown in Fig. 3 (top). Please see the Methods section 'Variational free energy' for further details. This specific form of variational free energy constitutes a class of cost functions for canonical neural networks, as we will see below.

In summary, variational free energy minimisation underwrites optimisation of posterior beliefs. In neurobiological formulations, it is usually assumed that neurons encode $\mathbf{s}_t$ and $\boldsymbol{\delta}_t$, while

**Variational free energy**

$$F = \sum_{\tau=1}^{t} \mathbf{s}_\tau \cdot \left( \ln \mathbf{s}_\tau - \ln \mathbf{A} \cdot o_\tau - \ln \mathbf{B}^\dagger \cdot \mathbf{s}_{\tau-1} - \ln D \right) + \sum_{\tau=1}^{t} \boldsymbol{\delta}_\tau \cdot \left( \ln \boldsymbol{\delta}_\tau - \left(1 - 2\Gamma_{t,\tau}\right) \ln \mathbf{C}^\dagger \cdot \mathbf{s}_{\tau-1} - \ln E \right) + \mathcal{O}(\ln t)$$

$$L = \int_0^t \begin{pmatrix} x(\tau) \\ \bar{x}(\tau) \end{pmatrix}^{\mathrm{T}} \left\{ \ln \begin{pmatrix} x(\tau) \\ \bar{x}(\tau) \end{pmatrix} - \ln \begin{pmatrix} \widehat{W}_1 & \overline{\widehat{W}_1} \\ \widehat{W}_0 & \overline{\widehat{W}_0} \end{pmatrix} \begin{pmatrix} o(\tau) \\ \bar{o}(\tau) \end{pmatrix} - \ln \begin{pmatrix} \hat{K}_1 & \overline{\hat{K}_1} \\ \hat{K}_0 & \overline{\hat{K}_0} \end{pmatrix} \begin{pmatrix} x(\tau - \Delta t) \\ \bar{x}(\tau - \Delta t) \end{pmatrix} - \begin{pmatrix} \phi_1 \\ \phi_0 \end{pmatrix} \right\} d\tau + \int_0^t \begin{pmatrix} y(\tau) \\ \bar{y}(\tau) \end{pmatrix}^{\mathrm{T}} \left\{ \ln \begin{pmatrix} y(\tau) \\ \bar{y}(\tau) \end{pmatrix} - (1 - 2\Gamma(t,\tau)) \ln \begin{pmatrix} \hat{V}_1 & \overline{\hat{V}_1} \\ \hat{V}_0 & \overline{\hat{V}_0} \end{pmatrix} \begin{pmatrix} x(\tau - \Delta t) \\ \bar{x}(\tau - \Delta t) \end{pmatrix} - \begin{pmatrix} \psi_1 \\ \psi_0 \end{pmatrix} \right\} d\tau + \mathcal{O}(1)$$

**Neural network cost function**

**Fig. 3 Mathematical equivalence between variational free energy and neural network cost functions, depicted by one-to-one correspondence of their components.** Top: variational free energy transformed from Eq. (5) using the Bayes theorem. Here, $\mathbf{B}^\dagger := \mathbf{B}^{\mathrm{T}} \mathrm{diag}[D]^{-1}$ and $\mathbf{C}^\dagger = \mathbf{C}^{\mathrm{T}} \mathrm{diag}[E]^{-1}$ indicate the inverse mappings, and $D$ and $E$ are the state and decision priors. Bottom: neural network cost function that is a counterpart to the aforementioned variational free energy. In this equation, $\widehat{W}_l := \mathrm{sig}(W_l)$, $\hat{K}_l := \mathrm{sig}(K_l)$, and $\hat{V}_l := \mathrm{sig}(V_l)$ (for $l = 0, 1$) indicate the sigmoid functions of synaptic strengths. Moreover, $\phi_l$ and $\psi_l$ are perturbation terms that characterise the bias in firing thresholds. Here, $\phi_l := \phi_l(W_l, K_l) = h_l - \ln \widehat{W}_l \vec{1} - \ln \hat{K}_l \vec{1}$ is a function of $W_l$ and $K_l$, while $\psi_l := \psi_l(V_l) = m_l - \ln \hat{V}_l \vec{1}$ is a function of $V_l$. When $\hat{\omega}_i := \mathrm{sig}(\omega_i)$ is the sigmoid function of $\omega_i$, $\omega_i \equiv \ln \hat{\omega}_i - \ln \overline{\hat{\omega}_i}$ holds for an arbitrary $\omega_i$. Using this relationship, Eq. (7) is transformed into the form presented at the bottom of this figure. This form of cost functions formally corresponds to variational free energy expressed on the top of this figure. Blue lines show one-to-one correspondence of their components.

synaptic strengths encode θ[17,18]. In what follows, we demonstrate that the internal states of canonical neural networks encode posterior beliefs.

**Canonical neural networks perform active inference.** In this section, we identify the neuronal substrates that correspond to components of the active inference scheme defined above. We consider a class of two-layer neural networks with recurrent connections in the middle layer (Fig. 1a). The modelling of the networks in this section (referred to as canonical neural networks) is based on the following three assumptions—that reflect physiological knowledge: (1) gradient descent on a cost function $L$ determines the updates of neural activity and synaptic weights (see Methods section 'Neural networks' for details); (2) neural activity is updated by the weighted sum of inputs, and its fixed point is expressed in a form of the sigmoid (or logistic) function; and (3) a modulatory factor mediates synaptic plasticity in a post hoc manner.

Based on assumption 2, we formulate neural activity in the middle layer ($x$) and output layer ($y$) as follows:

$$\begin{cases} \dot{x}(t) \propto -\underbrace{\mathrm{sig}^{-1}(x(t))}_{\text{leak current}} + \underbrace{(W_1 - W_0)o(t) + (K_1 - K_0)x(t - \Delta t)}_{\text{synaptic input}} + \underbrace{h_1 - h_0}_{\text{threshold}} \\ \dot{y}(t) \propto -\underbrace{\mathrm{sig}^{-1}(y(t))}_{\text{leak current}} + \underbrace{(V_1 - V_0)x(t - \Delta t)}_{\text{synaptic input}} + \underbrace{m_1 - m_0}_{\text{threshold}} \end{cases}$$

(6)

Here, $x(t) := (x_1(t), \ldots, x_{N_x}(t))^{\mathrm{T}}$ and $y(t) := (y_1(t), \ldots, y_{N_y}(t))^{\mathrm{T}}$ denote column vectors of firing intensities; $o(t) := (o_1(t), \ldots, o_{N_o}(t))^{\mathrm{T}}$ is a column vector of binary sensory inputs; $W_1, W_0 \in \mathbb{R}^{N_x \times N_o}$, $K_1, K_0 \in \mathbb{R}^{N_x \times N_x}$ and $V_1, V_0 \in \mathbb{R}^{N_y \times N_x}$ are synaptic strength matrices; and $h_1 := h_1(W_1, K_1)$, $h_0 := h_0(W_0, K_0)$, $m_1 := m_1(V_1)$ and $m_0 := m_0(V_0)$ are adaptive firing thresholds that depend on synaptic strengths. This model is derived as a reduction of a realistic neuron model through some approximations (see Supplementary Methods 3 for details).

One may think of $W_1$, $K_1$ and $V_1$ as excitatory synapses, whereas $W_0$, $K_0$ and $V_0$ can be regarded as inhibitory synapses. Here, $(W_1 - W_0)o(t)$ represents the total synaptic input from the sensory layer, and $(K_1 - K_0)x(t - \Delta t)$ forms a recurrent circuit with a time delay $\Delta t > 0$. Receiving inputs from the middle layer $x(t)$, the output-layer neural activity $y(t)$ determines the decision $\delta(t) := (\delta_1(t), \ldots, \delta_{N_\delta}(t))^{\mathrm{T}}$, that is, $\mathrm{Prob}[\delta_i(t) = 1] = y_i(t)$. We select the inverse sigmoid (i.e. logit) leak current to ensure that the fixed point of Eq. (6) (i.e. $x$ and $y$ that ensure $\dot{x} = 0$ and $\dot{y} = 0$) has the form of a sigmoid activation function (c.f.,

assumption 2). The sigmoid activation function is also known as the neurometric function[34].

Without loss of generality, Eq. (6) can be cast as the gradient descent on cost function $L$. Such a cost function can be identified by simply integrating the right-hand side of Eq. (6) with respect to $x$ and $y$, consistent with previous treatments[22]. Moreover, we presume that output-layer synapses ($V_1$, $V_0$) are updated through synaptic plasticity mediated by the modulator $\Gamma(t)$ (c.f., assumption 3; $0 \le \Gamma(t) \le 1$), as a model of plasticity modulations that are empirically observed[31–33]. Because neural activity and synaptic plasticity minimise the same cost function $L$, the derivatives of $L$ must generate the modulated synaptic plasticity. Under these constraints reflecting assumptions 1–3, a class of cost functions is identified as follows:

$$L = \int_0^t \begin{pmatrix} x(\tau) \\ \bar{x}(\tau) \end{pmatrix}^{\mathrm{T}} \left\{ \ln \begin{pmatrix} x(\tau) \\ \bar{x}(\tau) \end{pmatrix} - \begin{pmatrix} W_1 \\ W_0 \end{pmatrix} o(\tau) - \begin{pmatrix} K_1 \\ K_0 \end{pmatrix} x(\tau - \Delta t) - \begin{pmatrix} h_1 \\ h_0 \end{pmatrix} \right\} d\tau$$
$$+ \int_0^t \begin{pmatrix} y(\tau) \\ \bar{y}(\tau) \end{pmatrix}^{\mathrm{T}} \left\{ \ln \begin{pmatrix} y(\tau) \\ \bar{y}(\tau) \end{pmatrix} - (1 - 2\Gamma(t,\tau)) \begin{pmatrix} V_1 \\ V_0 \end{pmatrix} x(\tau - \Delta t) - \begin{pmatrix} m_1 \\ m_0 \end{pmatrix} \right\} d\tau$$
$$+ \mathcal{O}(1)$$

(7)

where $\bar{x}(\tau) := \vec{1} - x(\tau)$ with a vector of ones $\vec{1} = (1, \ldots, 1)^{\mathrm{T}}$. Here, $\mathcal{O}(1)$—that denotes a function of synaptic strengths—is of a smaller order than the other terms that are of order $t$. Thus, $\mathcal{O}(1)$ is negligible when $t$ is large. We suppose $\Gamma(t, \tau) = 0$ for $t - \Delta t < \tau \le t$ and $\Gamma(t, \tau) = \Gamma(t)$ for $0 \le \tau \le t - \Delta t$, to satisfy assumptions 1–3. This means that the optimisation of $L$ by associative plasticity is mediated by $\Gamma(t)$. We note that a gradient descent on $L$, i.e. $\dot{x} \propto -\mathrm{d}/\mathrm{d}t \cdot \partial L/\partial x$ and $\dot{y} \propto -\mathrm{d}/\mathrm{d}t \cdot \partial L/\partial y$, has the same functional form (and solution) as Eq. (6) (see Methods section 'Neural networks' and Supplementary Methods 4 for further details).

Synaptic plasticity rules conjugate to the above rate coding model can now be expressed as gradient descent on the same cost function $L$, according to assumption 1. To simplify notation, we define synaptic strength matrix as $\omega_i \in \{W_1, W_0, K_1, K_0, V_1, V_0\}$, presynaptic activity as $pre_i(t) \in \{o(t), o(t), x(t - \Delta t), x(t - \Delta t), x(t - \Delta t), x(t - \Delta t)\}$, postsynaptic activity as $post_i(t) \in \{x(t), \bar{x}(t), x(t), \bar{x}(t), y(t), \bar{y}(t)\}$ and firing thresholds as $n_i \in \{h_1, h_0, h_1, h_0, m_1, m_0\}$. Note that some variables (e.g. $x(t)$) appear several times because some synapses connect to the same pre- or postsynaptic neurons as other synapses. Thus, synaptic plasticity in the middle layer

**Table 2 Correspondence of variables and functions.**

| Neural network formation | | | Variational Bayes formation | |
|---|---|---|---|---|
| Sensory inputs | $o(t)$ | $\Longleftrightarrow$ | $o_t$ | Observations |
| Middle-layer neural activity | $\begin{pmatrix} x(t) \\ \bar{x}(t) \end{pmatrix}$ | $\Longleftrightarrow$ | $\mathbf{s}_t$ | State posterior |
| Output-layer neural activity | $\begin{pmatrix} y(t) \\ \bar{y}(t) \end{pmatrix}$ | $\Longleftrightarrow$ | $\boldsymbol{\delta}_t$ | Decision posterior |
| Feedback response | $\delta(t)$ | $\Longleftrightarrow$ | $\delta_t$ | Decision |
| Neuromodulator | $\Gamma(t)$ | $\Longleftrightarrow$ | $\Gamma_t$ | Risk function |
| Synaptic strengths | $W_l$ | $\Longleftrightarrow$ | $\mathrm{sig}^{-1}(\mathbf{A}_{1l})$ | Parameter posterior |
|  | $K_l$ | $\Longleftrightarrow$ | $\mathrm{sig}^{-1}(\mathbf{B}_{1l}^{\dagger})$ |  |
|  | $V_l$ | $\Longleftrightarrow$ | $\mathrm{sig}^{-1}(\mathbf{C}_{1l}^{\dagger})$ |  |
| Perturbation terms | $\phi := \begin{pmatrix} \phi_1 \\ \phi_0 \end{pmatrix}$ | $\Longleftrightarrow$ | $\ln D$ | State prior |
|  | $\psi := \begin{pmatrix} \psi_1 \\ \psi_0 \end{pmatrix}$ | $\Longleftrightarrow$ | $\ln E$ | Decision prior |
| Firing thresholds | $h_l$ | $\Longleftrightarrow$ | $\ln \mathbf{A}_{0l} \cdot \vec{1} + \ln \mathbf{B}_{0l}^{\dagger} \cdot \vec{1} + \ln D_l$ |  |
|  | $m_l$ | $\Longleftrightarrow$ | $\ln \mathbf{C}_{0l}^{\dagger} \cdot \vec{1} + \ln E_l$ |  |
| Initial synaptic strengths | $\lambda_l^W \odot \hat{W}_l^{\mathrm{init}}$ | $\Longleftrightarrow$ | $a_{1l}$ | Parameter prior |
|  | $\lambda_l^K \odot \hat{K}_l^{\mathrm{init}}$ | $\Longleftrightarrow$ | $b_{1l}$ |  |
|  | $\lambda_l^V \odot \hat{V}_l^{\mathrm{init}}$ | $\Longleftrightarrow$ | $c_{1l}$ |  |

Bold case variables (e.g. $\mathbf{s}_\tau$) denote the posterior expectations of the corresponding italic case random variables (e.g. $s_\tau$). Note that $W_l^{\mathrm{init}}, K_l^{\mathrm{init}}, V_l^{\mathrm{init}}$ are initial values of $W_l, K_l, V_l$ (for $l = 0, 1$) and $\lambda_l^W, \lambda_l^K, \lambda_l^V$ are inverse learning rate factors that express the insensitivity of synaptic strengths to plasticity. Please refer to the previous paper[22] for details.

$(i = 1, \ldots, 4)$ is derived as follows:

$$\dot{\omega}_i \propto -\frac{1}{t}\frac{\partial L}{\partial \omega_i} = \underbrace{\langle post_i(t)pre_i(t)^{\mathrm{T}} \rangle}_{\text{Hebbian plasticity}} + \underbrace{\langle post_i(t)\vec{1}^{\mathrm{T}} \rangle \odot \frac{\partial n_i}{\partial \omega_i}}_{\text{homeostatic plasticity}} \quad (8)$$

Moreover, synaptic plasticity in the output layer ($i = 5, 6$) is derived as follows:

$$\dot{\omega}_i \propto -\frac{1}{t}\frac{\partial L}{\partial \omega_i} = \underbrace{(1-2\Gamma(t))\langle post_i(t)pre_i(t)^{\mathrm{T}} \rangle}_{\text{modulated Hebbian plasticity}} + \underbrace{\langle post_i(t)\vec{1}^{\mathrm{T}} \rangle \odot \frac{\partial n_i}{\partial \omega_i}}_{\text{homeostatic plasticity}} \quad (9)$$

Here, $\langle post_i(t)pre_i(t)^{\mathrm{T}} \rangle := \frac{1}{t}\int_0^t post_i(\tau)pre_i(\tau)^{\mathrm{T}} d\tau$ indicates the average over time, $\odot$ indicates the element-wise product operator, and $t \gg \Delta t$.

These synaptic update rules are biologically plausible as they comprise Hebbian plasticity—determined by the outer product of pre- and postsynaptic activity—accompanied by an activity-dependent homoeostatic term. In Eq. (9), the neuromodulator $\Gamma(t)$—that encodes an arbitrary risk—alters the form of Hebbian plasticity in a post hoc manner. This can facilitate the association between past decisions and the current risk, thus leading to the optimisation of the decision rule to minimise future risk. In short, $\Gamma(t) < 0.5$ yields Hebbian plasticity, whereas $\Gamma(t) > 0.5$ yields anti-Hebbian plasticity. Empirical observations suggest that some modulators[31–33], such as dopamine neurons[35–37], are a possible neuronal substrate of $\Gamma(t)$; please see Discussion for further details.

Based on the above considerations, we now establish the formal correspondence between the neural network cost function and variational free energy. Under the aforementioned three minimal assumptions, we identify the neural network cost function as Eq. (7). Equation (7) can be transformed into the form shown in Fig. 3 (bottom) using sigmoid functions of synaptic strengths (e.g. $\hat{W}_l := \mathrm{sig}(W_l)$ for $l = 0, 1$). Here, the firing thresholds $(h_l, m_l)$ are replaced with the perturbation terms in the thresholds,

$\phi_l := h_l - \ln \overrightarrow{W_l}\vec{1} - \ln \overrightarrow{K_l}\vec{1}$ and $\psi_l := m_l - \ln \overrightarrow{V_l}\vec{1}$. Figure 3 depicts the formal equivalence between the neural network cost function (Fig. 3, bottom) and variational free energy (Fig. 3, top), visualised by one-by-one correspondence between their components. The components of variational free energy—including the log-likelihood function and complexities of states and decisions—re-emerge in the neural network cost function.

This means that when $(x(\tau)^{\mathrm{T}}, \bar{x}(\tau)^{\mathrm{T}})^{\mathrm{T}} = \mathbf{s}_\tau$, $(y(\tau)^{\mathrm{T}}, \bar{y}(\tau)^{\mathrm{T}})^{\mathrm{T}} = \boldsymbol{\delta}_\tau$, $\Gamma(t) = \Gamma_t$, $\hat{W}_l = \mathbf{A}_{1l}$, $\hat{K}_l = \mathbf{B}_{1l}^{\dagger}$, and $\hat{V}_l = \mathbf{C}_{1l}^{\dagger}$ (for $l = 0, 1$), the neural network cost function is identical to variational free energy, up to the negligible $\ln t$ residual. This further endorses the asymptotic equivalence of Eqs. (5) and (7).

The neural network cost function is characterised by the perturbation terms implicit in firing thresholds $\phi := (\phi_1^{\mathrm{T}}, \phi_0^{\mathrm{T}})^{\mathrm{T}}$ and $\psi := (\psi_1^{\mathrm{T}}, \psi_0^{\mathrm{T}})^{\mathrm{T}}$. These terms correspond to the state and decision priors, $\ln P(s_t) = \ln D = \phi$ and $\ln P(\delta_t) = \ln E = \psi$, respectively. Further, an arbitrary neuromodulator that regulates synaptic plasticity as depicted in Eq. (9) plays the role of the risk function in the POMDP model defined in Eq. (2), where the equivalence can be confirmed by comparing Eqs. (9) and (18). Hence, this class of cost functions for canonical neural networks is formally homologous to variational free energy, under the particular form of the POMDP generative model, defined in the previous section. In other words, Eqs. (2) and (5) express the class of generative models—and ensuing variational free energy—that ensure Eq. (1) is apt, for the class of canonical neural networks considered. We obtained this result based on analytic derivations —without reference to the complete class theorem—thereby confirming the proposition in Eq. (1). This in turn suggests that any canonical neural network in this class is implicitly performing active inference. Table 2 summarises the correspondence between the quantities of the neural network and their homologues in variational Bayes.

In summary, when a neural network minimises the cost function with respect to its activity and plasticity, the network self-organises to furnish responses that minimise a risk implicit in

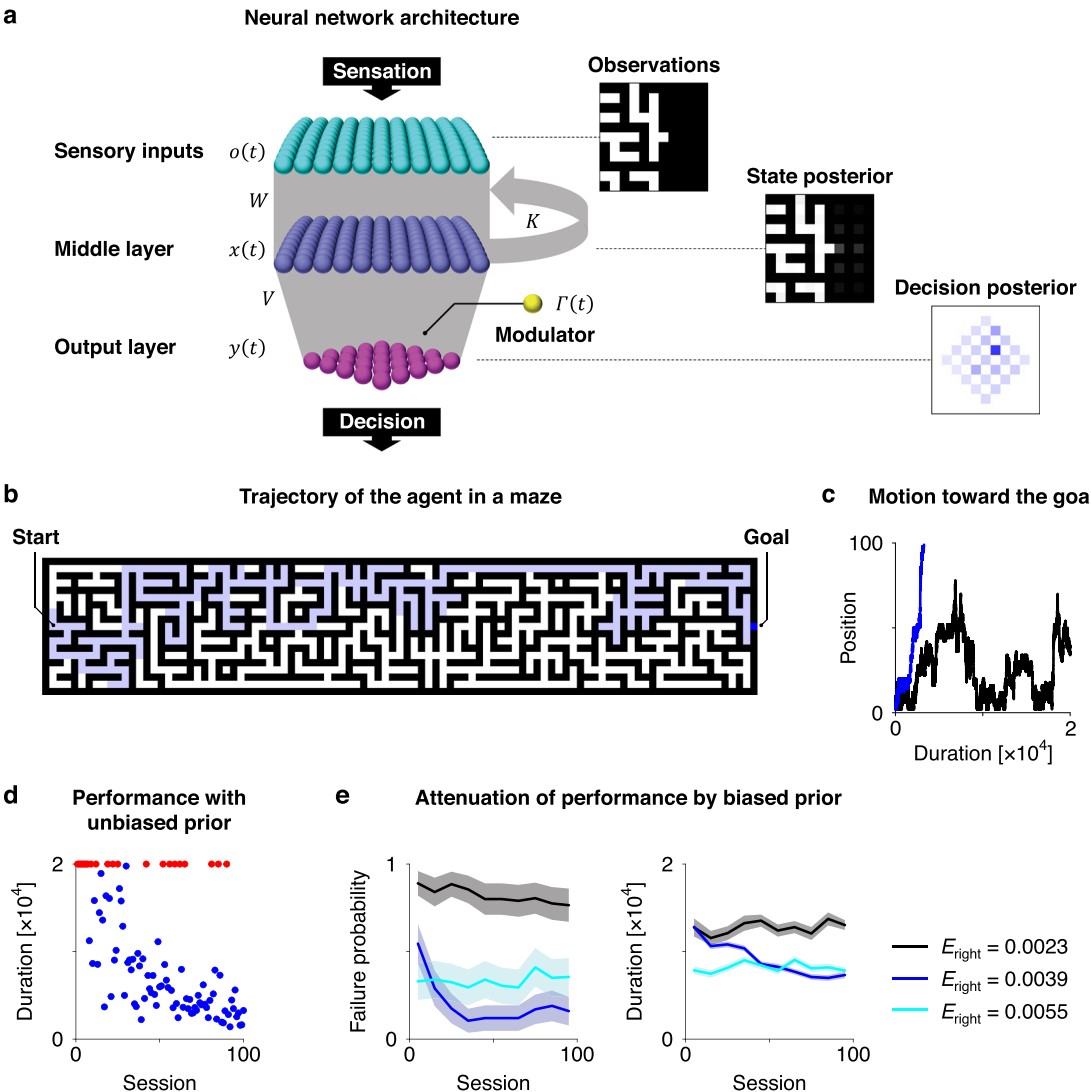

**Fig. 4 Simulations of neural networks solving maze tasks. a** Neural network architecture. The agent receives the states (pathway or wall) of the neighbouring 11 × 11 cells as sensory inputs. A decision here represents a four-step sequence of actions (selected from up, down, left or right), resulting in 256 options in total. The panels on the right depict observations and posterior beliefs about hidden states and decisions. **b** General view of the maze. The maze comprises a discrete state space, wherein white and black cells indicate pathways and walls, respectively. A thick blue cell indicates the current position of the agent, while the thin blue line is its trajectory. Starting from the left, the agent needs to reach the right edge of the maze within $T = 2 \times 10^4$ time steps. **c** Trajectories of the agent's x-axis position in sessions before (black, session 1) and after (blue, session 100) training. **d** Duration to reach the goal when the neural network operates under uniform decision priors $E_{\text{right}} = E_{\text{left}} = E_{\text{up}} = E_{\text{down}} = 1/256 \approx 0.0039$ (where $E_{\text{right}}$ indicates the prior probability to select a decision involving the rightward motion in the next step). Blue and red circles indicate succeeded and failed sessions, respectively. **e** Failure probability (left) and duration to reach the goal (right) when the neural network operates under three different prior conditions $E_{\text{right}} = 0.0023, 0.0039, 0.0055$ (black, blue and cyan, respectively), where $E_{\text{left}} = 0.0078 - E_{\text{right}}$ and $E_{\text{up}} = E_{\text{down}} = 0.0039$ hold. The line indicates the average of ten successive sessions. Although the neural network with $E_{\text{right}} = 0.0055$ exhibits better performance in the early stage, it turns out to overestimate a preference of the rightward motion in later stages, even when it approaches the wall. **e** was obtained with 20 distinct, randomly generated mazes. Shaded areas indicate the standard error. Refer to Methods section 'Simulations' for further details.

the cost function. This biological optimisation is identical to variational free energy minimisation under a particular form of the POMDP model. Hence, this equivalence indicates that minimising the expected risk through variational free energy minimisation is an inherent property of canonical neural networks featuring a delayed modulation of Hebbian plasticity.

**Numerical simulations**. Here, we demonstrate the performance of canonical neural networks using maze tasks—as an example of a delayed reward task. The agent comprised the aforementioned canonical neural networks (Fig. 4a). Thus, it implicitly performs

active inference by minimising variational free energy. The maze affords a discrete state space (Fig. 4b). The agent received the states of the neighbouring cells as sensory inputs, and its neural activity represented the hidden states (Fig. 4a, panels on the right; See Methods section 'Simulations' for further details). Although we denoted $s$ as hidden states, the likelihood mapping $A$ was a simple identity mapping in these simulations. When solving Eqs. (6), (8), and (9), the agent's neural network implicitly updates posterior beliefs about its behaviour based on the policy mapping. It then selects an appropriate action to move towards a neighbouring cell according to the inferred policy. The action was accepted if the selected movement was allowed.

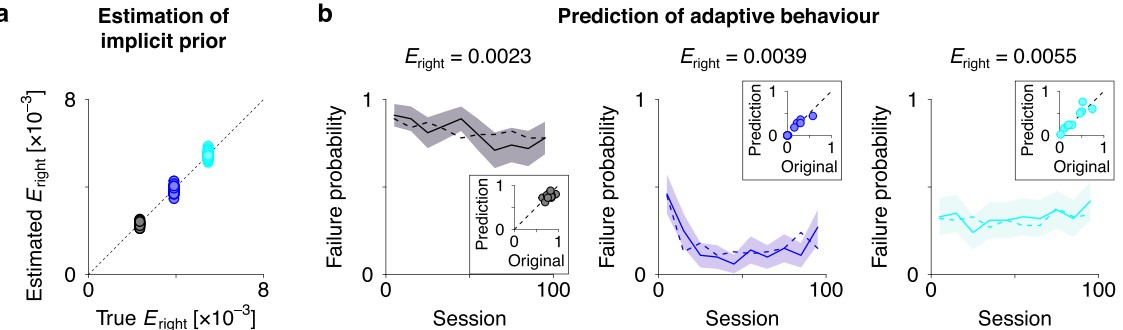

**Fig. 5 Estimation of implicit priors enables the prediction of subsequent learning. a** Estimation of implicit prior $E_{\text{right}}$—encoded by threshold factor $\psi$—under three different prior conditions (black, blue and cyan; c.f., Fig. 4). Here, $\psi$ was estimated through Bayesian inference based on sequences of neural activity, obtained with ten distinct mazes. Then, $E_{\text{right}}$ was computed by $\ln E_1 = \psi_1$ for each of 64 elements. The other 192 elements of $E_1$ (i.e. $E_{\text{left}}, E_{\text{up}}, E_{\text{down}}$) were also estimated. The sum of all the elements of $E_1$ was normalised to 1. **b** Prediction of the learning process within previously unexperienced, randomly generated mazes. Using the estimated $E$, we reconstructed the computational architecture (i.e. neural network) of the agent. Then, we simulated the adaptation process of the agent's behaviour using the reconstructed neural network and computed the trajectory of the probability of failure to reach the goal within $T = 2 \times 10^4$ time steps. The resulting learning trajectories (solid lines) predict the learning trajectories of the original agent (dashed lines) under three different prior conditions, in the absence of observed neural responses and behaviours. Lines and shaded areas indicate the mean and standard error, respectively. Inset panels depict comparisons between the failure probability of the original and reconstructed agent after learning (average over session 51–100), within ten previously unexperienced mazes. Refer to Methods section 'Data analysis' for further details.

Before training, the agent moved to a random direction in each step, resulting in a failure to reach the goal position (right end) within the time limit. During training, the neural network updated synaptic strengths depending on its neural activity and ensuing outcomes (i.e. risk). The training comprised a cycle of action and learning phases. In the action phase, the agent enacted a sequence of decisions, until it reached the goal or $T = 2 \times 10^4$ time steps passed (Fig. 4c). In the learning phase, the agent evaluated the risk associated with past decisions after a certain period: the risk was minimum (i.e. $\Gamma(t) = 0$) if the agent moved rightwards with a certain distance during the period; otherwise $\Gamma(t) = 0.45$ if the agent moved rightwards during the period, or $\Gamma(t) = 0.55$ if it did not. The synaptic strengths $V$ (i.e. the policy mapping) were then potentiated if the risk was low, or suppressed otherwise, based on Eq. (9). This mechanism made it possible to optimise decision making. Other synapses ($W, K$) were also updated based on Eq. (8), although we assumed a small learning rate to focus on the implicit policy learning. Through training, the neural network of the agent self-organised its behaviour to efficiently secure its goal (Fig. 4d). We also observed that modulations of Hebbian plasticity without delay did not lead to optimal behaviour, resulting in a considerably higher probability of failing to reach the goal. These results indicate that delayed modulation is essential to enable canonical neural networks to solve delayed reward tasks.

With this setup in place, we numerically validated the dependency of performance on the threshold factors ($\phi, \psi$). Consistent with our theoretical prediction—that $\phi$ and $\psi$ encode prior beliefs about hidden states ($D$) and decisions ($E$)—alternations of $\psi = \ln E$ from the optimum to a suboptimal value changed the landscape of the cost function (i.e. variational free energy), thereby providing suboptimal inferences and decisions (in relation to the environment). Subsequently, the suboptimal network firing thresholds led to a suboptimal behavioural strategy, taking a longer time or failing to reach the goal (Fig. 4e). Thus, we could attribute the agent's impaired performance to its suboptimal priors. This treatment renders neural activity and adaptive behaviours of the agent highly explainable and manipulatable in terms of the appropriate prior beliefs—implicit in firing thresholds—for a given task or environment. In other words, these results suggest that firing thresholds are the neuronal substrates that encode state and decision priors, as predicted mathematically.

Furthermore, when the updating of $\phi$ and $\psi$ is slow in relation to experimental observations, $\phi$ and $\psi$ can be estimated through Bayesian inference based on empirically observed neuronal responses (see Methods section 'Data analysis' for details). Using this approach, we estimated implicit prior $E$—which is encoded by $\psi$—from sequences of neural activity generated from the synthetic neural networks used in the simulations reported in Fig. 4. We confirmed that the estimator was a good approximation to the true $E$ (Fig. 5a). The estimation of $\phi$ and $\psi$ based on empirical observations offered the reconstruction of the cost function (i.e. variational free energy) that an agent employs. The resulting cost function could predict subsequent learning of behaviours within previously unexperienced, randomly generated mazes—without observing neural activity and behaviour (Fig. 5b). This is because—given the canonical neural network at hand—the learning self-organisation is based exclusively on state and decision priors, implicit in $\phi$ and $\psi$. Therefore, the identification of these implicit priors is sufficient to asymptotically determine the fixed point of synaptic strengths when $t$ is large (see Methods section 'Neural networks' for further details; see also ref. [22]). These results highlight the utility of the proposed equivalence to understand neuronal mechanisms underlying adaptation of neural activity and behaviour through accumulation of past experiences and ensuing outcomes.

## Discussion

Biological organisms formulate plans to minimise future risks. In this work, we captured this characteristic in biologically plausible terms under minimal assumptions. We derived simple differential equations that can be plausibly interpreted in terms of a neural network architecture that entails degrees of freedom with respect to certain free parameters (e.g. firing threshold). These free parameters play the role of prior beliefs in variational Bayesian formation. Thus, the accuracies of inferences and decisions depend upon prior beliefs, implicit in neural networks. Consequently, synaptic plasticity with false prior beliefs leads to suboptimal inferences and decisions for any task under consideration.

Based on the view of the brain as an agent that performs Bayesian inference, neuronal implementations of Bayesian belief updating have been proposed, which enables neural networks to store and recall spiking sequences[8], learn temporal dynamics and

causal hierarchy[9], extract hidden causes[10], solve maze tasks[11] and make plans to control robots[12]. In these approaches, the update rules are generally derived from Bayesian cost functions (e.g. variational free energy). However, the precise relationship between these update rules and the neural activity and plasticity of canonical neural networks has yet to be fully established.

We identified a one-to-one correspondence between neural network architecture and a specific POMDP implicit in that network. Equation (2) speaks to a unique POMDP model consistent with the neural network architecture defined in Eq. (6), where their correspondences are summarised in Table 2. This means that our scheme can be used to identify the form of POMDP, given an observable circuit structure. Moreover, the free parameters—that parameterise Eq. (6)—can be estimated using Eq. (24). This means that the generative model and ensuing variational free energy can, in principle, be reconstructed from empirical data. This offers a formal characterisation of implicit Bayesian models entailed by neural circuits, thereby enabling a prediction of subsequent learning. Our numerical simulations, accompanied by previous work[22], show that canonical neural networks behave systematically—and in a distinct way—depending on the implicit POMDP (Fig. 4). This kind of self-organisation depends on implicit prior beliefs, which can therefore be characterised empirically (Fig. 5).

A simple Hebbian plasticity strengthens synaptic wiring when pre- and post-synaptic neurons fire together, which enhances the association between (presynaptic) causes and (postsynaptic) consequences[28]. Hebbian plasticity depends on the activity level[29,30], spike timings[38,39] or burst timings[40] of pre- and post-synaptic neurons. Furthermore, modulatory factors can regulate the magnitude and parity of Hebbian plasticity, possibly with some time delay, leading to the emergence of various associative functions[31–33]. This means that neuromodulators can be read as encoding precision which regulates inference and learning[41]. These modulations have been observed empirically with various neuromodulators and neurotransmitters, such as dopamine[35–37,42,43], noradrenaline[44,45], muscarine[46] and GABA[47,48], as well as glial factors[49].

In particular, a delayed modulation of synaptic plasticity is well-known with dopamine neurons[35–37]. This speaks to a learning scheme that is conceptually distinct from standard reinforcement learning algorithms, such as the temporal difference learning with actor-critic models based on state-action value functions[3]. Please see the previous work[50] for a detailed comparison between active inference and reinforcement learning. Delayed modulations are also observed with noradrenaline and serotonin[51]. We mathematically demonstrated that such plasticity enhances the association between the pre-post mapping and the future value of the modulatory factor, where the latter is cast as a risk function. This means that postsynaptic neurons self-organise to react in a manner that minimises future risk. Crucially, this computation corresponds formally to variational Bayesian inference under a particular form of POMDP generative models, suggesting that the delayed modulation of Hebbian plasticity is a realisation of active inference. Regionally specific projections of neuromodulators may allow each brain region to optimise activity to minimise risk and leverage a hierarchical generative model implicit in cortical and subcortical hierarchies. This is reminiscent of theories of neuromodulation and (meta-)learning developed previously[52]. Our work may be potentially useful, when casting these theories in terms of generative models and variational free energy minimisation.

The complete class theorem[19–21] ensures that any neural network, whose activity and plasticity minimise the same cost function, can be cast as performing Bayesian inference. However, identifying the implicit generative model that underwrites any canonical neural network is a more delicate problem because the theorem does not specify a form of a generative model for a given canonical neural network. The posterior beliefs are largely shaped by prior beliefs, making it challenging to identify the generative model by simply observing systemic dynamics. To this end, it is necessary to commit to a particular form of the generative model and elucidate how the posterior beliefs are encoded or parameterised by the neural network states. This work addresses these issues by establishing a reverse-engineering approach to identify a generative model implicit in a canonical neural network, thereby establishing one-to-one correspondences between their components. Remarkably, a network of rate coding models with a sigmoid activation function formally corresponds to a class of POMDP models, which provide an analytically tractable example of the present equivalence (please refer to the previous paper[22] for further discussion).

It is remarkable that the proposed equivalence can be leveraged to identify a generative model that an arbitrary neural network implicitly employs. This contrasts with naive neural network models that address only the dynamics of neural activity and plasticity. If the generative model differs from the true generative process—that generates the sensory input—inferences and decisions are biased (i.e. suboptimal), relative to Bayes optimal inferences and decisions based on the right sort of prior beliefs. In general, the implicit priors may or may not be equal to the true priors; thus, a generic neural network is typically suboptimal. Nevertheless, these implicit priors can be optimised by updating free parameters (e.g. threshold factors $\phi, \psi$) based on the gradient descent on cost function $L$. By updating the free parameters, the network will eventually, in principle, becomes Bayes optimal for any given task. In essence, when the cost function is minimised with respect to neural activity, synaptic strengths and any other constants that characterise the cost function, the cost function becomes equivalent to variational free energy with the optimal prior beliefs. Simultaneously, the expected risk is minimised because variational free energy is minimised only when the precision of the risk ($\gamma_t$) is maximised (see Methods section 'Generative model' for further details).

When the rate coding activation function differs from the sigmoid function, it can be assumed that neurons encode state posteriors under a generative model that differs from a typical POMDP model considered in the main text (see Supplementary Methods 4 for details; see also ref. [22]). Nevertheless, the complete class theorem guarantees the existence of some pair of a generative model (i.e. priors) and cost function that corresponds to an arbitrary activation function. The form or time window of empirically observed plasticity rules can also be used to identify the implicit cost and risk functions—and further to reverse engineer the task or problem that the neural network is solving or learning: c.f., inverse reinforcement learning[53]. In short, neural activity and plasticity can be interpreted, universally, in terms of Bayesian belief updating.

The class of neural networks we consider can be viewed as a class of reservoir networks[54,55]. The proposed equivalence could render such reservoir networks explainable—and may provide the optimal plasticity rules for these networks to minimise future risk—by using the formal analogy to variational free energy minimisation (under the particular form of POMDP models). A clear interpretation of reservoir networks remains an important open issue in computational neuroscience and machine learning.

Assumption 1 places constraints on the relationship between neural activity and plasticity. The ensuing synaptic plasticity rules (Eqs. (8) and (9)) represent a key prediction of the proposed scheme. We have shown that these plasticity rules enable neural circuits to assimilate sensory inputs—as Bayes optimal encoders of external states—and generate responses, as Bayes optimal

controllers that minimise risk. Tracking changes in synaptic strengths is empirically difficult in large networks, and thus the functional form of synaptic plasticity is more difficult to establish, compared to that of synaptic responses. The current scheme enables one to predict the nature of plasticity, given observed neural (and behavioural) responses, under ideal Bayesian assumptions. This is sometimes referred to as meta-Bayesian inference[56]. The validity of these predictions can, therefore, in principle, be verified using empirical measures of neural activity and behaviour. In short, the current scheme predicts specific plasticity rules for canonical neural networks, when learning a given task.

The equivalence between neural network dynamics and gradient flows on variational free energy is empirically testable using electrophysiological recordings or functional imaging of brain activity. For instance, neuronal populations in layers 2/3 and 5 of the parietal lobe of rodents have been shown to encode posterior expectations about hidden sound cues, which are used to reach a goal in uncertain environments[57]. According to the current formulation, synaptic afferents to these expectation-coding populations—from neurons encoding sensory information—should obey the plasticity rule in Eq. (8) and self-organise to express the likelihood mapping (A). Related predictions are summarised in Table 2. We have previously shown that the self-organisation of in vitro neural networks minimises empirically computed variational free energy in a manner consistent with variational free energy minimisation under a POMDP generative model[58,59]. Our analyses in the present work speak to the predictive validity of the proposed formulation: when the threshold factors $(\phi, \psi)$ can be treated as constants—during a short experiment—we obtain the analytical form of fixed points for synaptic update rules (Methods section 'Neural networks'). Furthermore, $\phi$ and $\psi$ can be estimated using empirical data (Methods section 'Data analysis'). This approach enables the reconstruction of the cost function and prediction of subsequent learning process, as demonstrated in Fig. 5 using in silico data. Hence, it is possible to examine the predictive validity of the proposed theory by comparing the predicted synaptic trajectory with the actual trajectory. In future work, we hope to address these issues using in vitro and in vivo data.

Crucially, the proposed equivalence guarantees that an arbitrary neural network that minimises its cost function—possibly implemented in biological organisms or neuromorphic hardware[60,61]—can be cast as performing variational Bayesian inference. Thus, in addition to contributions to neurobiology, this notion can dramatically reduce the complexity of designing self-learning neuromorphic hardware to perform various types of tasks; therefore, it offers a simple architecture and low computational cost. This leads to a unified design principle for biologically inspired machines such as neuromorphic hardware to perform statistically optimal inference, learning, prediction, planning, and decision making.

In summary, a class of biologically plausible cost functions for canonical neural networks can be cast as variational free energy. Formal correspondences exist between priors, posteriors and cost functions. This means that canonical neural networks that optimise their cost functions implicitly perform active inference. This approach enables identification of the implicit generative model and reconstruction of variational free energy that neural networks employ. This means that, in principle, neural activity, behaviour and learning through plasticity can be predicted under Bayes optimality assumptions.

## Methods
**Generative model**. The proposed POMDP model comprises $N_s$-dimensional hidden states $s_t \in \{0, 1\}^{N_s}$ that depend on the previous states $s_{t-1}$ through a

transition probability of $B^\delta$, and a process of generating $N_o$-dimensional observations $o_t \in \{0, 1\}^{N_o}$ from those states through a likelihood mapping $A$ (Fig. 2). Here, the transition probability $B^\delta$ is a function of $N_\delta$-dimensional decisions of an agent $\delta_t \in \{0, 1\}^{N_\delta}$, indicating that the agent's behaviour changes the subsequent states of the external milieu. Each state, observation, and decision take the values 1 or 0. We use $o_{1:t} = \{o_1, \ldots, o_t\}$ to denote a sequence of observations. Hereafter, $i$ indicates the $i$-th observation, $j$ indicates the $j$-th hidden state and $k$ indicates the $k$-th decision.

Due to the multidimensional (i.e. factorial) nature of the states, $A$, $B$ and $C$ are usually the outer products of submatrices (i.e. tensors); see also ref. [22]. The probability of an observation is determined by the likelihood mapping, from $s_t$ to $o_t^{(i)}$, in terms of a categorical distribution: $P(o_t^{(i)}|s_t, A^{(i)}) = \mathrm{Cat}(A^{(i)})$, where the elements of $A^{(i)}$ are given by $A_{1\vec{l}}^{(i)} = P(o_t^{(i)} = 1|s_t = \vec{l}, A^{(i)})$ and $A_{0\vec{l}}^{(i)} = \vec{1} - A_{1\vec{l}}^{(i)}$ (see also Table 1). This encodes the probability that $o_t^{(i)}$ takes 1 or 0 when $s_t = \vec{l} = (l_1, \ldots, l_N)^T \in \{0, 1\}^{N_s}$. The hidden states are determined by the transition probability, from $s_{t-1}$ to $s_t$, depending on a given decision, in terms of a categorical distribution: $P(s_t^{(j)}|s_{t-1}, B^{\delta(j)}) = \mathrm{Cat}(B^{\delta(j)})$. As defined in Eq. (3), a generative model of decisions $\delta_\tau$ is conditioned on the current risk $\gamma_t \in \{0, 1\}$ that obeys $P(\gamma_t) = \mathrm{Cat}((\Gamma_t, \overline{\Gamma_t})^T)$, where $\overline{\Gamma_t} = 1 - \Gamma_t$. This can be viewed as a postdiction of past decisions. For $1 \le \tau \le t - 1$, the conditional probability of $\delta_\tau^{(k)}$ has a form of a mixture model: $P(\delta_\tau^{(k)}|s_{\tau-1}, \gamma_t, C^{(k)}) = \mathrm{Cat}(C^{(k)})^{\overline{\gamma_t}} \mathrm{Cat}(C^{(k)} \oslash C^{(k)})^{\gamma_t}$. Conversely, for $\tau = t$, $\delta_t^{(k)}$ is sampled from $P(\delta_t^{(k)}|s_{t-1}, \gamma_t, C^{(k)}) = \mathrm{Cat}(C^{(k)})$ to minimise the future risk because the agent does not yet observe the consequence of the current decision.

Equation (3) represents a mechanism by which the agent learns to select the preferred decisions. This is achieved by updating the policy mapping $C$ depending on the consequences of previous decisions. Following Eq. (3), when observing $\gamma_t = 0$, the agent regards that the past decisions were sampled from the preferable policy mapping $C$ that minimises $\Gamma_t$, and thereby updates the posterior belief of $C$ to facilitate the association between $\delta_\tau$ and $s_{\tau-1}$. In contrast, when observing $\gamma_t = 1$, the agent hypothesises that this is because the past decisions were sampled from the unpreferable policy mapping, and thereby updates the posterior belief of $C$ to reduce (forget) the association. In short, this postdiction evaluates the past decisions after observing their consequence. The posterior belief of $C$ is then updated by associating the past decision rule (policy) and current risk, leading to the optimisation of decisions to minimise future risk. Interestingly, this behavioural optimisation mechanism derived from the Bayesian inference turns out to be identical to a post hoc modulation of Hebbian plasticity (see Eqs. (7) and (9), and Methods section 'Neural networks').

An advantage of this generative model—based on counterfactual causality—is that the agent does not need to explicitly compute the expected future risk based on the current states, because it instead updates the policy mapping $C$, by associating the current risk with past decisions. Note that this construction of risk corresponds to a simplification of expected free energy, that would normally include risk and ambiguity, where risk corresponds to the Kullback–Leibler divergence between the posterior predictive and prior distribution over outcomes[17,18]. However, by using a precise likelihood mapping, ambiguity can be discounted and expected free energy reduces to the sort of decision risk considered in this work. In other words, one can consider that the expected free energy is implicit in the policy mapping $C$. From this perspective, the risk $\Gamma_t$ is associated with precision that regulates the magnitude of expected free energy; refer to Supplementary Methods 2 for further details.

We can now define the generative model as Eq. (2), where $P(s_1|s_0, \delta_0, B) = P(s_1) = \mathrm{Cat}(D)$ and $P(\delta_1|s_0, \gamma_t, C) = P(\delta_1) = \mathrm{Cat}(E)$ are assumed. We further suppose that $\delta_\tau$, given $s_{\tau-1}$, is conditionally independent of $\{o_\tau, s_\tau\}$, and that only the generation of $\delta_\tau$ depends on $\gamma_t$, as visualised in the factor graph (Fig. 2). Equation (2) can be further expanded as follows:

$$P(o_{1:t}, s_{1:t}, \delta_{1:t}, \gamma_t, \theta) = P(\gamma_t) \cdot \prod_{i=1}^{N_o} P(A^{(i)}) \cdot \prod_{j=1}^{N_s} P(B^{\delta(j)}) \cdot \prod_{k=1}^{N_\delta} P(C^{(k)})$$
$$\cdot \prod_{\tau=1}^{t} \left\{ \prod_{i=1}^{N_o} P(o_\tau^{(i)}|s_\tau, A^{(i)}) \cdot \prod_{j=1}^{N_s} P(s_\tau^{(j)}|s_{\tau-1}, \delta_{\tau-1}, B^{\delta(j)}) \right.$$
$$\left. \cdot \prod_{k=1}^{N_\delta} P(\delta_\tau^{(k)}|s_{\tau-1}, \gamma_t, C^{(k)}) \right\}$$

(10)

The prior beliefs of $A_{\cdot\vec{l}}^{(i)}$, $B_{\cdot\vec{l}}^{\delta(j)}$, and $C_{\cdot\vec{l}}^{(k)}$ are defined by Dirichlet distributions $P(A_{\cdot\vec{l}}^{(i)}) = \mathrm{Dir}(a_{\cdot\vec{l}}^{(i)})$, $P(B_{\cdot\vec{l}}^{\delta(j)}) = \mathrm{Dir}(b_{\cdot\vec{l}}^{\delta(j)})$ and $P(C_{\cdot\vec{l}}^{(k)}) = \mathrm{Dir}(c_{\cdot\vec{l}}^{(k)})$ with concentration parameters $a_{\cdot\vec{l}}^{(i)}$, $b_{\cdot\vec{l}}^{\delta(j)}$ and $c_{\cdot\vec{l}}^{(k)}$, respectively. As described in the Results section, this form of the generative model is suitable to characterise a class of canonical neural networks defined by Eq. (6). This means that none of the aforementioned assumptions—regarding the generative model—limit the scope of the proposed equivalence between neural networks and variational Bayesian inference, as long as neural networks satisfy assumptions 1–3.

**Variational free energy**. The agent aims to minimise surprise, or equivalently maximise the marginal likelihood of outcomes, by minimising variational free energy as a tractable proxy. Thereby, they perform approximate or variational Bayesian inference. From the above-defined generative model, we motivate a mean-field approximation to the posterior distribution as follows:

$$Q(s_{1:t}, \delta_{1:t}, \theta) = Q(\theta)Q(s_{1:t})Q(\delta_{1:t}) = Q(A)Q(B)Q(C)\prod_{\tau=1}^{t}Q(s_\tau)Q(\delta_\tau) \quad (11)$$

Here, the posterior beliefs of $s_\tau$ and $\delta_\tau$ are categorical distributions, $Q(s_\tau) = \mathrm{Cat}(\mathbf{s}_\tau)$ and $Q(\delta_\tau) = \mathrm{Cat}(\boldsymbol{\delta}_\tau)$, respectively. Whereas, the posterior beliefs of $A$, $B$ and $C$ are Dirichlet distributions, $Q(A) = \mathrm{Dir}(\mathbf{a})$, $Q(B) = \mathrm{Dir}(\mathbf{b})$ and $Q(C) = \mathrm{Dir}(\mathbf{c})$, respectively. In this expression, $\mathbf{s}_\tau$ and $\boldsymbol{\delta}_\tau$ represent the expectations between 0 and 1, and $\mathbf{a}$, $\mathbf{b}$ and $\mathbf{c}$ express the (positive) concentration parameters.

In this paper, the posterior transition mapping is averaged over all possible decisions, $\mathbf{B} = \mathrm{E}_{Q(\delta)}[\mathbf{B}^\delta]$, to ensure exact correspondence to canonical neural networks. Moreover, we suppose that $\mathbf{A}$ comprises the outer product of submatrices $\mathbf{A}^{(i,j)} \in \mathbb{R}^{2\times2}$ to simplify the calculation of the posterior beliefs, i.e. $\mathbf{A}_{l\cdot}^{(i)} = \mathbf{A}_{l\cdot}^{(i,1)} \otimes \cdots \otimes \mathbf{A}_{l\cdot}^{(i,N_s)}$ for $l = 0, 1$. We also suppose that $\mathbf{B}$ and $\mathbf{C}$ comprise the outer products of submatrices $\mathbf{B}^{(j,j')} \in \mathbb{R}^{2\times2}$ and $\mathbf{C}^{(k,j)} \in \mathbb{R}^{2\times2}$, respectively. The expectation over the parameter posterior $Q(\theta) = \prod_{i=1}^{N_o}\prod_{j=1}^{N_s}Q(A^{(i,j)})\cdot$

$\prod_{j=1}^{N_s}\prod_{j'=1}^{N_s}Q(B^{(j,j')})\cdot\prod_{k=1}^{N_\delta}\prod_{j=1}^{N_s}Q(C^{(k,j)})$ is denoted as $\mathrm{E}_{Q(\theta)}[\cdot] := \int \cdot Q(\theta)d\theta$. Using this, the posterior expectation of a parameter $\Theta \in \{A^{(i,j)}, B^{(j,j')}, C^{(k,j)}\}$ is expressed using the corresponding concentration parameter $\theta \in \{\mathbf{a}^{(i,j)}, \mathbf{b}^{(j,j')}, \mathbf{c}^{(k,j)}\}$ as follows:

$$\begin{cases} \boldsymbol{\Theta}_{\cdot l} := \mathrm{E}_{Q(\Theta_{\cdot l})}[\Theta_{\cdot l}] = \boldsymbol{\theta}_{\cdot l} \oslash (\boldsymbol{\theta}_{1l} + \boldsymbol{\theta}_{0l}) \\ \ln\boldsymbol{\Theta}_{\cdot l} := \mathrm{E}_{Q(\Theta_{\cdot l})}[\ln\Theta_{\cdot l}] = \psi(\boldsymbol{\theta}_{\cdot l}) - \psi(\boldsymbol{\theta}_{1l} + \boldsymbol{\theta}_{0l}) = \ln(\boldsymbol{\theta}_{\cdot l} \oslash (\boldsymbol{\theta}_{1l} + \boldsymbol{\theta}_{0l})) + \mathcal{O}((\boldsymbol{\theta}_{1l} + \boldsymbol{\theta}_{0l})^{-1}) \end{cases}$$
(12)

for $l = 0, 1$, where $\psi(\cdot)$ is the digamma function.

In terms of decisions, because $P(\delta_\tau|s_{\tau-1}, C, \gamma_t = 1) = \mathrm{Cat}(C'\oslash C)$ in this setup, the complexity associated with past decision is given by $\mathcal{D}_{\mathrm{KL}}[Q(\delta_\tau)||P(\delta_\tau|s_{\tau-1}, \gamma_t, C)] = \mathrm{E}_{Q(\delta_\tau)Q(s_{\tau-1})Q(C)P(\gamma_t)}[\ln Q(\delta_\tau) - \ln P(\delta_\tau|s_{\tau-1}, \gamma_t, C)] = \boldsymbol{\delta}_\tau \cdot \{\ln\boldsymbol{\delta}_\tau - (1 - \Gamma_t)\ln\mathbf{C}\mathbf{s}_{\tau-1} + \Gamma_t\ln\mathbf{C}\mathbf{s}_{\tau-1}\}$ for $1 \leq \tau \leq t-1$, up to the $C'$-dependent term which is negligible when computing the posterior beliefs. Whereas, the current decision is made to minimise the complexity $\mathcal{D}_{\mathrm{KL}}[Q(\delta_t)||P(\delta_t|s_{t-1}, C)] = \boldsymbol{\delta}_t \cdot (\ln\boldsymbol{\delta}_t - \ln\mathbf{C}\mathbf{s}_{t-1})$.

Variational free energy is defined as a functional of the posterior beliefs, given as:

$$F(o_{1:t}, Q(s_{1:t}, \delta_{1:t}, \theta)) := \mathrm{E}_{Q(s_{1:t}, \delta_{1:t}, \theta)P(\gamma_t)}[-\ln P(o_{1:t}, \delta_{1:t}, s_{1:t}, \gamma_t, \theta) + \ln Q(s_{1:t}, \delta_{1:t}, \theta)]$$
$$= \sum_{\tau=1}^{t}\mathrm{E}_{Q(s_\tau)Q(s_{\tau-1})Q(A)Q(B)}[\ln Q(s_\tau) - \ln P(o_\tau|s_\tau, A) - \ln P(s_\tau|s_{\tau-1}, \delta_{\tau-1}, B^\delta)]$$
$$+ \sum_{\tau=1}^{t}\mathrm{E}_{Q(\delta_\tau)Q(s_{\tau-1})Q(C)P(\gamma_t)}[\ln Q(\delta_\tau) - \ln P(\delta_\tau|s_{\tau-1}, \gamma_t, C)]$$
$$+ \mathcal{D}_{\mathrm{KL}}[P(A)||Q(A)] + \mathcal{D}_{\mathrm{KL}}[P(B)||Q(B)] + \mathcal{D}_{\mathrm{KL}}[P(C)||Q(C)] + H[\gamma_t]$$
(13)

This provides an upper bound of sensory surprise $-\ln P(o_{1:t})$. Here, $\mathcal{D}_{\mathrm{KL}}[\cdot||\cdot]$ is the complexity of parameters scored by the Kullback–Leibler divergence. Minimisation of variational free energy is attained when the entropy of the risk, $H[\gamma_t] := \mathrm{E}_{P(\gamma_t)}[-\ln P(\gamma_t)] = -\Gamma_t\ln\Gamma_t - \overline{\Gamma_t}\ln\overline{\Gamma_t}$, is minimised. This is achieved when $\Gamma_t$ shifts toward 0, meaning that the risk minimisation is a corollary of variational free energy minimisation (the case where $\Gamma_t$ shifts toward 1 is negligible). Under the POMDP scheme, $F$ is expressed as a function of the posterior expectations, $F = F(o_{1:t}, \mathbf{s}_{1:t}, \boldsymbol{\delta}_{1:t}, \theta)$. Thus, using the vector expression, variational free energy under our POMDP model is provided as follows:

$$F(o_{1:t}, \mathbf{s}_{1:t}, \boldsymbol{\delta}_{1:t}, \theta) = \underbrace{\sum_{\tau=1}^{t}\mathbf{s}_\tau\cdot(\ln\mathbf{s}_\tau - \ln\mathbf{A}\cdot o_\tau - \ln\mathbf{B}\mathbf{s}_{\tau-1})}_{\text{accuracy+state complexity}}$$
$$+ \underbrace{\sum_{\tau=1}^{t}\boldsymbol{\delta}_\tau\cdot(\ln\boldsymbol{\delta}_\tau - (1-2\Gamma_{t,\tau})\ln\mathbf{C}\mathbf{s}_{\tau-1})}_{\text{decision complexity}}$$
$$+ \underbrace{(\mathbf{a}-a)\cdot\ln\mathbf{A} - \ln\mathcal{B}(\mathbf{a}) + (\mathbf{b}-b)\cdot\ln\mathbf{B} - \ln\mathcal{B}(\mathbf{b}) + (\mathbf{c}-c)\cdot\ln\mathbf{C} - \ln\mathcal{B}(\mathbf{c})}_{\text{parameter complexity}}$$
(14)

Here, $\mathcal{B}(\mathbf{a}_{\cdot l}) \equiv \Gamma(\mathbf{a}_{1l})\Gamma(\mathbf{a}_{0l})/\Gamma(\mathbf{a}_{1l} + \mathbf{a}_{0l})$ is the beta function (where $\Gamma(\cdot)$ is the gamma function), and $\ln\mathbf{A}\cdot o_\tau$ indicates the inner product of $\ln\mathbf{A}$ and one-hot expressed $o_\tau$, which is a custom to express the sum of the product of $(\ln\mathbf{A}^{(i,j)})^\mathrm{T} \in \mathbb{R}^{2\times2}$ and $(o_\tau^{(i)}, \overline{o_\tau^{(i)}})^\mathrm{T} \in \mathbb{R}^2$ over all $i$.

The first and second terms of Eq. (14)—comprising accuracy and the complexity of state and decision—increases in proportion to time $t$. Conversely, other terms—the complexity of parameters—increases in the order of $\ln t$, which is thus negligible when $t$ is large. Hence, we will drop the parameter complexity by assuming that the scheme has experienced a sufficient number of observations. Please see the previous work[22] for further details. The entropy of the risk is omitted as it does not explicitly appear in the inference.

Based on the Bayes theorem, $P(s_\tau|s_{\tau-1}, B^\delta) \propto P(s_{\tau-1}|s_\tau, B^\delta)P(s_\tau)$ and $P(\delta_\tau|s_{\tau-1}, \gamma_t, C) \propto P(s_{\tau-1}|\delta_\tau, \gamma_t, \theta)P(\delta_\tau)$ hold, where $P(\delta_\tau)$ is supposed to be a flat prior belief. Thus, the inverse transition and policy mappings are given as $\mathbf{B}^\dagger = \mathbf{B}^\mathrm{T}\mathrm{diag}[D]^{-1}$ and $\mathbf{C}^\dagger = \mathbf{C}^\mathrm{T}\mathrm{diag}[E]^{-1}$, respectively. Thus, $\mathbf{s}_\tau\cdot\ln\mathbf{B}\mathbf{s}_{\tau-1} = \mathbf{s}_\tau\cdot(\ln\mathbf{B}^\dagger\cdot\mathbf{s}_{\tau-1} + \ln D)$ and $\boldsymbol{\delta}_\tau\cdot(1 - 2\Gamma_{t,\tau})\ln\mathbf{C}\mathbf{s}_{\tau-1} = \boldsymbol{\delta}_\tau\cdot((1 - 2\Gamma_{t,\tau})\ln\mathbf{C}^\dagger\cdot\mathbf{s}_{\tau-1} + \ln E)$ hold. Accordingly, Eq. (14) becomes

$$F(o_{1:t}, \mathbf{s}_{1:t}, \boldsymbol{\delta}_{1:t}, \theta) = \sum_{\tau=1}^{t}\mathbf{s}_\tau\cdot(\ln\mathbf{s}_\tau - \ln\mathbf{A}\cdot o_\tau - \ln\mathbf{B}^\dagger\cdot\mathbf{s}_{\tau-1} - \ln D)$$
$$+ \sum_{\tau=1}^{t}\boldsymbol{\delta}_\tau\cdot(\ln\boldsymbol{\delta}_\tau - (1-2\Gamma_{t,\tau})\ln\mathbf{C}^\dagger\cdot\mathbf{s}_{\tau-1} - \ln E) + \mathcal{O}(\ln t)$$
(15)

as expressed in Fig. 3 (top). Here, the prior beliefs about states and decisions, $P(s_\tau) = \mathrm{Cat}(D)$ and $P(\delta_\tau) = \mathrm{Cat}(E)$, alter the landscape of variational free energy. We will see below that this specific form of variational free energy corresponds formally to a class of cost functions for canonical neural networks.

**Inference and learning**. Inference optimises the posterior beliefs about the hidden states and decisions by minimising variational free energy. The posterior beliefs are updated by the gradient descent on $F$, $\dot{\mathbf{s}}_t \propto -\partial F/\partial\mathbf{s}_t$ and $\dot{\boldsymbol{\delta}}_t \propto -\partial F/\partial\boldsymbol{\delta}_t$. The fixed point of these updates furnishes the posterior beliefs, which are analytically computed by solving $\partial F/\partial\mathbf{s}_t = 0$ and $\partial F/\partial\boldsymbol{\delta}_t = 0$. Thus, from Eq. (15), the posterior belief about the hidden states is provided as follows:

$$\mathbf{s}_t = \sigma(\ln\mathbf{A}\cdot o_t + \ln\mathbf{B}^\dagger\cdot\mathbf{s}_{t-1} + \ln D) \quad (16)$$

Moreover, the posterior belief about the decisions is provided as follows:

$$\boldsymbol{\delta}_t = \sigma(\ln\mathbf{C}^\dagger\cdot\mathbf{s}_{t-1} + \ln E) \quad (17)$$

Here, $\sigma(\cdot)$ denotes the softmax function; and $D$ and $E$ denote the prior beliefs about hidden states and decisions, respectively, which we assume are fixed in this paper. Note that Eqs. (16) and (17) are equivalent to $\mathbf{s}_t = \sigma(\ln\mathbf{A}\cdot o_t + \ln\mathbf{B}\mathbf{s}_{t-1})$ and $\boldsymbol{\delta}_t = \sigma(\ln\mathbf{C}\mathbf{s}_{t-1})$, respectively, as $\mathbf{B}^\dagger = \mathbf{B}^\mathrm{T}\mathrm{diag}[D]^{-1}$ and $\mathbf{C}^\dagger = \mathbf{C}^\mathrm{T}\mathrm{diag}[E]^{-1}$. Notably, $\mathbf{s}_t = (\mathbf{s}_{t1}^\mathrm{T}, \mathbf{s}_{t0}^\mathrm{T})^\mathrm{T} = (s_{t1}^{(1)}, \ldots, s_{t1}^{(N_s)}, s_{t0}^{(1)}, \ldots, s_{t0}^{(N_s)})^\mathrm{T}$ indicates a block column vector of the state posterior under a mean-field assumption, where $\mathbf{s}_{t1}^{(i)}$ is the posterior belief that $s_t^{(i)}$ takes a value of 1. Because $s_t^{(i)}$ takes a binary value, $\mathbf{s}_{t1}^{(i)}$ has the form of a sigmoid function. Here, we assume that only the state posterior $\mathbf{s}_t$ at the latest time is updated at each time $t$; thus, no message pass exists from $\mathbf{s}_{t+1}$ to $\mathbf{s}_t$. The state posterior at time $t-1$ is retained in the previous value. This treatment corresponds to the Bayesian filter, as opposed to the smoother usually considered in active inference schemes.

Equations (16) and (17) are analogue to a two-layer neural network that entails recurrent connections in the middle layer. In this analogy, $\mathbf{s}_{t1}$ and $\boldsymbol{\delta}_{t1}$ are viewed as the middle- and output-layer neural activity, respectively. Moreover, $\ln\mathbf{A}\cdot o_t$, $\ln\mathbf{B}^\dagger\cdot\mathbf{s}_{t-1}$, and $\ln\mathbf{C}^\dagger\cdot\mathbf{s}_{t-1}$ correspond to synaptic inputs, and $\ln D$ and $\ln E$ relate to firing thresholds. These priors and posteriors turn out to be identical to the components of canonical neural networks, as described in the Results and Methods section 'Neural networks'.

Furthermore, learning optimises the posterior beliefs about the parameters $\theta = \{\mathbf{a}, \mathbf{b}, \mathbf{c}\}$ by minimising variational free energy. The posterior beliefs are updated by the gradient descent on $F$, $\dot{\theta} \propto -\partial F/\partial\theta$. By solving the fixed point $\partial F/\partial\theta = O$ of Eq. (14), the posterior beliefs about parameters are provided as follows:

$$\begin{cases} \mathbf{a} = a + \sum_{\tau=1}^{t}o_\tau\otimes\mathbf{s}_\tau = t\langle o_t\otimes\mathbf{s}_t\rangle + \mathcal{O}(1) \\ \mathbf{b} = b + \sum_{\tau=1}^{t}\mathbf{s}_\tau\otimes\mathbf{s}_{\tau-1} = t\langle\mathbf{s}_t\otimes\mathbf{s}_{t-1}\rangle + \mathcal{O}(1) \\ \mathbf{c} = c + \sum_{\tau=1}^{t-1}(1-2\Gamma_t)\boldsymbol{\delta}_\tau\otimes\mathbf{s}_{\tau-1} + \boldsymbol{\delta}_t\otimes\mathbf{s}_{t-1} = t\langle(1-2\Gamma_t)\langle\boldsymbol{\delta}_t\otimes\mathbf{s}_{t-1}\rangle\rangle + \mathcal{O}(1) \end{cases}$$
(18)

Note that $\otimes$ denotes the outer product operator, and $\langle\cdot\rangle$ indicates the average over time, e.g. $\langle o_t\otimes\mathbf{s}_t\rangle := \frac{1}{t}\sum_{\tau=1}^{t}o_\tau\otimes\mathbf{s}_\tau$. These posterior beliefs are usually obtained as an average over multiple sessions to ensure convergence. Here, $a, b, c$ are the prior beliefs, which are of order 1 and thus negligibly small relative to the leading order term when $t$ is large. Thus, the posterior expectations of any parameters $\Theta$ (i.e. $\boldsymbol{\Theta}$ and $\ln\boldsymbol{\Theta}$) are obtained using Eqs. (12) and (18). These parameter posteriors turn out to correspond formally to synaptic strengths $(W, K, V)$ owing to the equivalence of variational free energy and neural network cost function.

**Neural networks**. In this section, we elaborate the one-to-one correspondences between components of canonical neural networks and variational Bayes, via an analytically tractable model. Neurons respond quickly to a continuous stimulus stream with a timescale faster than typical changes in sensory input. For instance, a peak of spiking responses in the visual cortex (V1) appears between 50 and 80 ms after a visual stimulation[62,63], which is substantially faster than the temporal autocorrelation timescale of natural image sequences (~500 ms)[64,65]. Thus, we

consider the case where the neural activity converges to a fixed point, given a sensory stimulus. We note that the present equivalence is derived from the differential equation (Eq. (6)), but not from its fixed point; thus, the equivalence holds true irrespective of the time constant of neurons. To rephrase, neural networks with a large time constant formally correspond to Bayesian belief updating with a large time constant, which implies an insufficient convergence of the posterior beliefs.

Updates of neural activity are defined by Eq. (6). When the neural activity asymptotically converges to a steady state, the fixed point of Eq. (6)—i.e., $x$ and $y$ that give $\dot{x} = 0$ and $\dot{y} = 0$—is provided as follows:

$$\begin{cases} x(t) = \text{sig}((W_1 - W_0)o(t) + (K_1 - K_0)x(t - \Delta t) + h_1 - h_0) \\ y(t) = \text{sig}((V_1 - V_0)x(t - \Delta t) + m_1 - m_0) \end{cases} \quad (19)$$

The adaptive firing thresholds are given as functions of synaptic strengths, $h_l = \ln \overrightarrow{W_l \vec{1}} + \ln \overrightarrow{K_l \vec{1}} + \phi_l$ and $m_l = \ln \overrightarrow{V_l \vec{1}} + \psi_l$ (for $l = 0, 1$), where $\exp(\phi_1) + \exp(\phi_0) = \vec{1}$ and $\exp(\psi_1) + \exp(\psi_0) = \vec{1}$ hold in the element-wise sense. The form of Eq. (19) is identical to the state and decision posteriors in Eqs. (16) and (17) of the variational Bayesian formation. Further details are provided in Supplementary Methods 5.

Considering that neural activity corresponds to the posterior beliefs about states and decisions, one might consider the relationship between synaptic strengths and parameter posteriors. As we mathematically demonstrated in the Results section, owing to the equivalence between variational free energy $F$ and the neural network cost function $L$, i.e. Eq. (5) versus Eq. (7), synaptic strengths correspond formally to parameter posteriors. The ensuing synaptic update rules—derived as the gradient descent on $L$—are expressed in Eqs. (8) and (9). They have a biologically plausible form, comprising Hebbian plasticity accompanied by an activity-dependent homoeostatic term. The product of $\Gamma(t)$ and the associated term modulates plasticity depending on the quality of past decisions, after observing their consequences, leading to minimisation of the future risk. In other words, the modulation by $\Gamma(t)$ represents a postdiction that the agent implicitly conducts, wherein the agent regards its past decisions as preferable when $\Gamma(t)$ is low and memorises the strategy. Conversely, it regards them as unpreferable when $\Gamma(t)$ is high and forgets the strategy.

In particular, when $\phi$ and $\psi$ are constants, the fixed point of synaptic strengths that minimise $L$ is expressed analytically as follows: for simplification, we employ the notation using $\omega_i$, $pre_i$, $post_i$ and $n_i$, as described in the Results section. The derivative of firing threshold $n_i$ with respect to synaptic strength matrix $\omega_i$ yields the sigmoid function of $\omega_i$, i.e. $\partial n_i / \partial \omega_i = -\text{sig}(\omega_i)$. The fixed point of synaptic strengths ensures $\partial L / \partial \omega_i = O$. Thus, from Eqs. (8) and (9), it is analytically expressed as

$$\omega_i = \text{sig}^{-1}\left( \left\langle post_i(t)pre_i(t)^{\text{T}} \right\rangle \oslash \left\langle post_i(t)\overrightarrow{\vec{1}}^{\text{T}} \right\rangle \right) \quad (20)$$

for $i = 1, \dots, 4$, and

$$\omega_i = \text{sig}^{-1}\left( \left\langle (1 - 2\Gamma(t))\left\langle post_i(t)pre_i(t)^{\text{T}} \right\rangle \right\rangle \oslash \left\langle post_i(t)\overrightarrow{\vec{1}}^{\text{T}} \right\rangle \right) \quad (21)$$

for $i = 5, 6$ using the element-wise division $\oslash$. They are obtained as an average over multiple sessions. Equations (20) and (21) correspond to the posterior belief about parameters $A, B, C$, which are shown in Eq. (18).

We note that when neural activity and plasticity follow differential equations, without loss of generality, we can consider a common cost function for a sufficiently long training time $t$. When neural activity and plasticity are expressed as

$$\begin{cases} \dot{x}(t) \propto f'(x(t), o(t), W) \\ \dot{W} \propto \frac{1}{t}\int_0^t g'(x(\tau), o(\tau), W)d\tau \end{cases} \quad (22)$$

a cost function of the following form is implied:

$$L = -\int_{t-\Delta t}^t f(x(\tau), o(\tau), W)d\tau - \int_0^{t-\Delta t} g(x(\tau), o(\tau), W)d\tau \quad (23)$$

From this cost function, one can derive gradient descent rules for both neural activity and plasticity in the limit of a large $t$. This owes to the different time scales of updating $x(t)$ and $W$, known as an adiabatic approximation. Therefore, for a sufficiently large $t$, the existence of cost functions that satisfy assumption 1 is a mathematical truism. It should be emphasised that unless $f$ matches $g$, the neural and synaptic dynamics result in biased inference and learning; therefore, the cost function defined in Eq. (7) is apt.

**Simulations.** In Fig. 4, the neural network of the agent was characterised by a set of internal states $\varphi = \{x, y, W, K, V, \phi, \psi\}$, where $W$ means the set of $W_1$ and $W_0$. Neural activity $(x, y)$ was updated by following Eq. (6), while synaptic strengths $(W, K, V)$ were updated by following Eqs. (8) and (9). Here, we supposed that neural activity converges quickly to the steady state relative to the change of observations. This treatment allowed us to compute the network dynamics based on Eqs. (19–21), which could reduce the computational cost for numerical simulations. Synaptic strengths $W$ were initialised as a matrix sufficiently close to the identity matrix; whereas, synaptic strengths $K$ and $V$ were initialised as matrices with uniform values. This treatment served to focus on the policy learning implicit in

the update of $V$. The threshold factors $(\phi, \psi)$, which encoded the prior beliefs about hidden states $(D)$ and decisions $(E)$, were predefined and fixed over the sessions. In Fig. 4e, we varied $E$ to show how performance depends on these priors. Namely, $E_1 = (E_{\text{right}}, \dots, E_{\text{right}}, E_{\text{left}}, \dots, E_{\text{left}}, E_{\text{up}}, \dots, E_{\text{up}}, E_{\text{down}}, \dots, E_{\text{down}})^{\text{T}} \in [0, 1]^{256}$ was characterised by four values $E_{\text{right}}, E_{\text{left}}, E_{\text{up}}, E_{\text{down}} \in [0, 1]$, where $E_{\text{right}}$ indicates the prior probability to select a decision involving the rightward motion in the next step.

**Data analysis.** When the belief updating of implicit priors $(D, E)$ is slow in relation to experimental observations, $D$ and $E$ (which are encoded by $\phi$ and $\psi$) can be viewed as being fixed over a short period of time, as an analogy to homoeostatic plasticity over longer time scales[66]. Thus, $\phi$ and $\psi$ can be statistically estimated by a conventional Bayesian inference (or maximum likelihood estimation, given a flat prior) based on empirically observed neuronal responses. In this case, the estimators of $\phi$ and $\psi$ are obtained as follows:

$$\begin{cases} \phi = \ln\left( \dfrac{\langle x(t) \rangle}{\langle \bar{x}(t) \rangle} \right) \\ \psi = \ln\left( \dfrac{\langle y(t) \rangle}{\langle \bar{y}(t) \rangle} \right) \end{cases} \quad (24)$$

Note that we suppose constraints $\exp(\phi_1) + \exp(\phi_0) = \vec{1}$ and $\exp(\psi_1) + \exp(\psi_0) = \vec{1}$. This assumption finesses the estimation of implicit priors (Fig. 5a)—owing to the equivalence $\phi = \ln D$ and $\psi = \ln E$ (see Fig. 3 and Table 2) —and the ensuing reconstruction of variational free energy, which was conducted by substituting Eq. (24) into Eq. (15). Equation (24) indicates that the average activity encodes prior beliefs, which is consistent with the experimental observations; in that the activity of sensory areas reflects prior beliefs[67].

Furthermore, given a generative model, the posterior beliefs of external states were updated based on a sequence of sensory inputs by minimising variational free energy, without reference to neural activity or behaviour. Thus, the reconstructed variational free energy enabled us to predict subsequent inference and learning of the agent, without observing neural activity or behaviour (Fig. 5b). In Fig. 5, for simplicity, the form of the risk function was supposed to be known when reconstructing the cost function. Although we did not estimate $\phi$ in Fig. 5, the previous work showed that our approach can estimate $\phi$ from simulated neural activity data[22].

**Reporting summary.** Further information on research design is available in the Nature Research Reporting Summary linked to this article.

## Data availability

All relevant data are within the paper. Figures 4 and 5 were generated using the authors' scripts (see Code Availability).

## Code availability

The MATLAB scripts are available at GitHub https://github.com/takuyaisomura/reverse_engineering or Zenodo https://doi.org/10.5281/zenodo.5748850[68]. The scripts are covered under the GNU General Public License v3.0.

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

## Acknowledgements

We are grateful to Hitoshi Okamoto, Makio Torigoe and Lancelot Da Costa for discussions. This work was supported in part by the grant of Joint Research by the National Institutes of Natural Sciences (NINS Programme No. 01112005, 01112102). T.I. is funded by RIKEN Center for Brain Science. H.S. is funded by JSPS KAKENHI Grant Number JP 20K11709, 21H05246 and New Energy and Industrial Technology Development Organization (NEDO), Japan. K.J.F. is funded by a Wellcome Principal Research Fellowship (Ref: 088130/Z/09/Z). The funders had no role in study design, data collection and analysis, decision to publish, or preparation of the manuscript.

## Author contributions

Conceptualisation, T.I., H.S., and K.J.F.; Formal analysis, Simulation and Writing—original draft T.I.; Writing—review & editing, T.I., H.S. and K.J.F. All authors made

substantial contributions to conception, design and writing of the article. All authors have read and agreed to the published version of the manuscript.

## Competing interests

The authors declare no competing interests.
