## [Transparent Peer Review File · Communications Biology]

This manuscript has been previously reviewed at another Nature Portfolio journal. This document only contains reviewer comments and rebuttal letters for versions considered at Communications Biology.

4 July 2021

Dear Editors,

Thank you for handling this manuscript and sending us the reviewers' constructive comments. We have revised the manuscript and have added Supplementary Methods based on their comments as described below. We hope that these revisions are sufficient to meet your and the reviewers' expectations.

Sincerely yours,

Takuya Isomura

Reviewer #1 (Remarks to the Author):

The paper by Isomura and colleagues presents a mathematical analysis of a canonical neural circuit, showing that it implements free energy minimization under a partially observable Markov decision process.

Overall, I found this to be a clearly written and rigorous paper, although very dense and hard to keep track of all the notation (probably prohibitively so for the uninitiated reader). The equivalence will be interesting to people working on active inference.

[Response]

We would like to thank the reviewer for their effort in reviewing and commenting on our manuscript. In particular, we appreciate the comments about this work being interesting to people working on active inference. Since the active inference offers a unifying perspective on perception and action, we envisage that the manuscript will attract readers in the broad areas of biology. For example, a preprint version of this manuscript, posted on 11 Dec 2020, has an altmetric of 19. As detailed below, we have revised the manuscript according to your suggestions. We hope that these changes are satisfactory.

However, my main concern is that this paper may be of less relevance to a broader audience, because there is very little engagement with actual biology. What data does this model help explain? What are its testable predictions? The authors point to the observation that dopamine modulates synaptic plasticity, but the same set of findings is invoked by other theories (e.g., conventional reinforcement learning theories). The authors also make some very general points about testability but don't talk about any specific experiments (either existing or possible).

[Response]

Thank you for pointing out these issues regarding the engagement with biology. We have tried to revise the manuscript to address the reviewer's concerns. We provide point-by-point revisions dealing with (i) explanation of data and prediction by the model, (ii) the novelty of the theoretical interpretation of dopaminergic modulation and how it differs from reinforcement learning, and (iii) specific experiments.

(i) Explanation of data and prediction by the model

We agree with the importance of explaining the results from a theoretical view and provide testable predictions. To further strengthen these aspects, we have added the following to the Discussion and Methods sections:

“Furthermore, modulatory factors can regulate the magnitude and parity of Hebbian plasticity, possibly with some time delay, leading to the emergence of various associative functions [26–28]. This means that neuromodulators can be read as encoding precision which regulates inference and learning [36].”

“Assumption 1 places constraints on the relationship between neural activity and plasticity. The ensuing synaptic plasticity rules (equations (8) and (9)) represent a key prediction of the proposed scheme. We have shown that these plasticity rules enable neural circuits to assimilate sensory inputs—as Bayes optimal encoders of external states—and generate responses, as Bayes optimal controllers that minimise risk. Tracking changes in synaptic strengths is empirically difficult in large networks, and thus the functional form of synaptic plasticity is more difficult to establish, compared to that of synaptic responses. The current scheme enables one to predict the nature of plasticity, given observed neural (and behavioural) responses, under ideal Bayesian assumptions. This is sometimes referred to as meta-Bayesian inference [52]. The validity of these predictions can therefore, in principle, be verified using empirical measures of neural activity and behaviour. In short, the current scheme predicts specific plasticity rules for canonical neural networks, when learning a given task.”

“Equation (24) indicates that the average activity encodes prior beliefs, which is consistent with experimental observations that the activity of sensory areas reflects prior beliefs [60].”

We would be happy to extend this discussion if biological plausibility and predictions remain a particular concern of the reviewers.

(ii) The novelty of the theoretical interpretation of dopaminergic modulation and how it differs from reinforcement learning

In terms of the reviewer's concern that other theories cover our formulation of dopaminergic modulation of synaptic plasticity (e.g., conventional reinforcement learning theories), we emphasise that the ‘delayed modulation’ of Hebbian plasticity (Yagishita et al., 2014; He et al., 2015; Wieland et al., 2015; Brzosko et al., 2017) is essential to enable canonical neural networks to ‘plan’. This notion differs from standard reinforcement learning theories such as temporal

difference learning and actor-critic models based on state value functions. To address the reviewer’s concern, we have emphasised this, in the Results and Discussion sections:

“We also observed that modulations of Hebbian plasticity without delay did not lead to optimal behaviour, resulting in a considerably higher probability of failing to reach the goal. These results indicate that delayed modulation is essential to enable canonical neural networks to solve delayed reward tasks.”

“In particular, a delayed modulation of synaptic plasticity is well-known with dopamine neurons [30–32]. This speaks to a learning scheme that is conceptually distinct from standard reinforcement learning algorithms, such as the temporal difference learning with actor-critic models based on state-action value functions [45]. Please see [46] for a detailed comparison between active inference and reinforcement learning. Delayed modulations are also observed with noradrenaline and serotonin [47].”

Moreover, we show that, although some heuristic approaches have provided synaptic update rules—for standard neural networks—that increase expected reward, they are not guaranteed to find the optimal solution because of the inherent difficulty in analysing standard neural networks. In contrast, our approach provides the optimal neural and synaptic update rules to render the output Bayes optimal. Namely, these rules minimise the risk associated with the future outcomes. This follows from the formal analogy between standard neural networks and variational Bayesian inference we consider. This is important in understanding efficient and plausible learning rules for standard neural networks.

As evidence for this, we show that canonical neural networks—featuring delayed modulation of Hebbian plasticity—exhibit a smaller failure probability and a shorter duration to reach the goal in the maze task shown in **Fig. 4**, compared to standard reinforcement learning approaches based on actor-critic models.

Fig. R1. Comparisons with reinforcement learning.

Here, the red dot reports the result of the proposed method (obtained with 20 mazes), while the blue dots indicate the results of actor-critic models with 5 different learning rates. Error bars

denote the standard error. We would be happy to include this comparison as a supplementary figure—to emphasise an advantage of the proposed scheme—if the reviewer thinks it is useful.

Finally, we note that the biological plausibility of active inference has been established to a much greater degree than in deep learning and reinforcement learning. On our reading, there is an increasing move away from reinforcement learning towards active inference in several domains (e.g., in the IEEE community, machine learning, and emerging initiatives in the industry). In light of this, the current paper may be of particular interest to people who are currently working on reinforcement learning—because it shows that standard neural networks can be configured to perform active inference.

(iii) Specific experiments

To emphasise specific experiments to test the theoretical predictions, we have added the following to the Discussion section:

“For instance, neuronal populations in layers 2/3 and 5 of the parietal lobe of rodents have been shown to encode posterior expectations about hidden sound cues, which are used to reach a goal in uncertain environments [53]. According to the current formulation, synaptic afferents to these expectation-coding populations—from neurons encoding sensory information—should obey the plasticity rule in equation (8) and self-organise to express the likelihood mapping (A). Related predictions are summarised in **Table 1.**”

I agree with the authors that it's interesting to identify the implicit generative model of the canonical neural network as a POMDP. Many environments can be modeled as POMDPs (most tasks used in experimental neuroscience and psychology could be modeled this way). So an important question is whether we can establish a *specific* implied POMDP that the network is optimizing for a given task. The network should behave systematically differently across tasks depending on the implied POMDP, which should provide some experimental leverage to test the model's predictions.

[Response]

While we appreciate the points made by the reviewer, we believe that the reviewer overlooked our main analytical result. We have established a procedure to explicitly identify a *specific* POMDP—that the network is implicitly optimising—for a wide range of task setups. Please note that equation (2) speaks to a *specific* and *unique* POMDP model consistent with the neural network architecture defined in equation (6), where their correspondences are summarised in Table 1. To clarify this, we have inserted the following paragraph in the Discussion section to emphasise this point:

“We identified a one-to-one correspondence between a neural network architecture and a specific POMDP implicit in that network. Equation (2) speaks to a unique POMDP model consistent with the neural network architecture defined in equation (6), where their correspondences are

summarised in **Table 1**. This means that our scheme can be used to identify the form of POMDP, given an observable circuit structure. Moreover, the free parameters that characterise equation (6) can be estimated using equation (24). This means that the generative model and ensuing variational free energy can—in principle—be reconstructed from empirical data. This offers a formal characterisation of implicit Bayesian models entailed by neural circuits, thereby enabling a prediction of subsequent learning. Our numerical simulations, accompanied by previous work [17], show that canonical neural networks behave systematically—and in a distinct way—depending on the implicit POMDP (**Fig. 4**). This kind of self-organisation depends on implicit prior beliefs, which can therefore be characterised empirically (**Fig. 5**).”

Other comments:

I'm not exactly sure what the relevant of the complete class theorem is here. While it is true that all admissible decision rules correspond to some variational inference procedure, it is not the case that the correspondence holds for *approximate* variational inference (e.g., in the mean-field approximation used here).

[Response]

Technically, the complete class theorem can apply to both approximate and exact Bayesian inference. It is a generic and fundamental principle that may be usefully unpacked for the reinforcement and deep learning community. To address this point more carefully, we added the following descriptions on the complete class theorem as Supplementary Methods, and now refer to it in the Results section:

“1. Notes on the complete class theorem

Here, we highlight several technical points with regards to the complete class theorem [14–16]. When the internal states φ^* minimise a cost function, there are no other internal states φ that can reduce the cost further; thus, we have $L(o_{1:t}, \varphi^*) \leq L(o_{1:t}, \varphi)$ for any φ . This means that φ^* meets the definition of an admissible decision rule. Therefore, according to the complete class theorem, φ^* can be cast as the Bayes optimal solution under, at least, a pair of Bayesian cost function and prior beliefs.

Moreover, the variational free energy minimisation framework covers both approximate and exact Bayesian inference. Indeed, minimisation of variational free energy without (mean-field) approximation simply yields Bayes’ theorem. Hence, equation (1) holds true for any generative model with prior beliefs. Variational free energy minimisation—under a mean-field approximation—is admissible when the generative model (including the mean-field assumption) corresponds to the generative process that generates sensory data. This is asymptotically the case when the hidden states are generated in a mutually independent manner, and the time scales for updating states and parameters are sufficiently separable. In short, the complete class theorem applies to both approximate and exact Bayesian inference.”

Moreover, in the section “Overview of equivalence between neural networks and variational Bayes”, the Bayesian solution is not necessary to be an approximate one; thus, we have revised the text as follows:

“Here, $Q(\vartheta)$ approximates (or possibly exactly equals to) $P(\vartheta|o_{1:t})$.”

Finally, we emphasise that our main analytical results in the subsequent Results and Methods sections are based on analytic derivations without reference to the complete class theorem. Thus, our analyses rather support our proposition in equation (1). We have emphasised this in the Results section:

“We obtained this result based on analytic derivations—without reference to the complete class theorem—thereby confirming the proposition in equation (1).”

p. 3: May want to clarify that variational free energy is the negative of the ELBO.

p. 9: "define generative model" -> "define a generative model"

p. 24: "set up" -> "setup"

[Response]

Thank you. We have fixed these typos.

Reviewer #2 (Remarks to the Author):

The authors present a demonstration that a class of 'canonical neural networks' which can be found to be equivalent to discrete-state-space active inference with a POMDP generative model. The claim is that we can thus interpret this class of recurrent neural network models as performing bayesian (variational) inference and that with this equivalence we can interpret biological factors such as variable firing rate thresholds as Bayesian priors. Secondly, if we consider the canonical neural network interacting with the environment using the output of its final layer as actions or 'decisions' then we can interpret its actions as minimizing future risk. The core goal of trying to propose an explicit generative model for a biological system in terms of active inference -- effectively 'reverse-engineering' the bayesian nature of a specific dynamical system

To my knowledge the claims in this paper are novel and correct. It is also interesting that a class of neural network models can be so closely related to the active inference paradigm.

[Response]

We would like to thank the reviewer for these gracious and valuable comments. As described below, we have revised the manuscript to address the issues raised. We hope these revisions are what you and the editor had in mind.

One concern I have is about the generality and utility of the results. Specifically, it is not clear to me that the 'canonical' neural network presented here is actually representative of the kind of neural network models found in neuroscience. It is to some extent, as it has hebbian-ish learning rules and a linear summation of input dynamics -- however the only nonlinearity is due to the inverse sigmoid leak term, which I believe is nonstandard. While this has a fixed point with the standard form of a nonlinear rate-coded neuron, the actual dynamics do not follow this pattern. Moreover, where this nonlinearity comes from is not clear to me, since it does not appear in the objective function. Because of this, I think the generality of the results would be much improved if the authors expanded upon the neural validity of their canonical neural network, and especially linked to other works which utilize them. Making it clear that this class of 'canonical neural networks' has independent interest outside of active inference is important, and that if the authors wish to engage with other literatures and communities, a more thorough description of how this class relates to other models in the neuroscience literature would be a good step towards this.

[Response]

Thank you for highlighting this issue. We have added the following as a section in Supplementary Methods and cited it in the Results and Discussion sections:

“3. Derivation of rate coding models

Without loss of generality, a recurrent neural network model with nonlinear synaptic inputs

$$u(t) = g\left((W_1 - W_0)o(t) + (K_1 - K_0)\phi(u(t - \Delta t)) + h_1 - h_0\right)$$

can be rewritten using an auxiliary variable $x(t) := \phi(u(t))$ as follows

$$x(t) = g'\left((W_1 - W_0)o(t) + (K_1 - K_0)x(t - \Delta t) + h_1 - h_0\right)$$

where $g' := \phi \circ g$. Thus, the differential equation for neural dynamics considered in this work

$$\dot{x}(t) \propto -f(x(t)) + (W_1 - W_0)o(t) + (K_1 - K_0)x(t - \Delta t) + h_1 - h_0$$

is sufficient to describe canonical neural network models. Although this work focuses on the case wherein $f(x(t)) = \text{sig}^{-1}(x(t))$ —because this case is optimal when the external state space is discrete—previous work has derived biologically plausible cost functions for canonical neural networks with a general form for $f(x(t))$ [17]. Please see also the Discussion, where we revisit these more general cases.

Concerning other models in neuroscience, widely used neural activity models—such as the FitzHugh-Nagumo model [61,62] and the Hindmarsh–Rose model [63]—update the activity $x(t)$ via linear summation of a nonlinear transformation of $x(t)$ and synaptic inputs precisely in the same manner as in the formulation we consider. In particular, the inverse sigmoid function approximates the cubic nonlinearity of the FitzHugh-Nagumo model. Thus, these models can be viewed as a family of canonical neural networks.

In term of the derivation of the nonlinear activation function, the neural network cost function

involves the integral of $f(x(t))$ by construction. The first term of equation (7) is

$$L = \int_0^t \left(\frac{x(\tau)}{\bar{x}(\tau)} \right)^T \left\{ \ln \left(\frac{x(\tau)}{\bar{x}(\tau)} \right) - \begin{pmatrix} W_1 \\ W_0 \end{pmatrix} o(\tau) - \begin{pmatrix} K_1 \\ K_0 \end{pmatrix} x(\tau - \Delta t) - \begin{pmatrix} h_1 \\ h_0 \end{pmatrix} \right\} d\tau$$

Because $\bar{x}(\tau) \equiv \vec{1} - x(\tau)$, the derivative of the first part yields

$$\frac{\partial}{\partial x} \left\{ \left(\frac{x(\tau)}{\bar{x}(\tau)} \right)^T \ln \left(\frac{x(\tau)}{\bar{x}(\tau)} \right) \right\} = \ln x(\tau) - \ln \bar{x}(\tau) = \ln \frac{x(\tau)}{\vec{1} - x(\tau)} = \text{sig}^{-1}(x(\tau))$$

which is known as the inverse sigmoid (or logit) function. Here, $\vec{1}$ is a vector of ones. Thus, the gradient descent with respect to $x(t)$ yields equation (6) as follows:

$$\dot{x}(t) \propto -\frac{d}{dt} \frac{\partial L}{\partial x} = -\text{sig}^{-1}(x(t)) + (W_1 - W_0)o(t) + (K_1 - K_0)x(t - \Delta t) + h_1 - h_0$$

Note that this holds true when a general nonlinear form of $f(x(t))$ is considered [17].”

I also have some questions about the relationship of the active inference scheme presented here compared to the more standard paradigm as presented in (Friston 2017, a neural process theory). Two core differences immediately stand out. The first is that this scheme optimizes actions with respect to the variational free energy (which is the cost function) instead of the expected free energy, as is done in the standard active inference paradigm, where the variational free energy is only used for perception and the softmax of the expected free energy is used for action selection and planning.

[Response]

Thank you for highlighting this interesting issue. We have added the following as Supplementary Methods and have cited it in the Results and Methods sections:

“2. Comparisons with the standard active inference formulation

Here, we discuss the relationship between the active inference scheme considered in this work and the standard paradigm [13]. The current scheme optimises actions or decisions with respect to variational free energy (F), whereas in standard active inference, variational free energy is used for perception while the expected free energy (G) is used for action selection and planning. In this setting, expected free energy furnishes prior beliefs over policies:

$$P(\pi) = e^{-\gamma \cdot G(\pi)}$$

Here, γ denotes a precision. Hence, the expected free energy is the component of variational free energy that is used to optimise posterior beliefs about hidden states *and* policies. In general formulations [13], a policy π indicates a sequence of actions, and variational free energy is defined as a functional of the posterior over $\vartheta = \{s_{1:t}, \pi, \theta\}$ as follows:

$$\begin{aligned}
F(o_{1:t}, \boldsymbol{\vartheta}) &= -\mathbb{E}_Q[\ln P(o_{1:t}|\vartheta)] + \mathcal{D}_{\text{KL}}[Q(\vartheta)||P(\vartheta)] \\
&= \sum_{\tau=1}^t \mathbb{E}_Q[F(\tau, \pi)] + \mathcal{D}_{\text{KL}}[Q(\pi)||P(\pi)] + \frac{\mathcal{D}_{\text{KL}}[Q(\theta)||P(\theta)]}{\mathcal{O}(\ln t)} \\
&= \boldsymbol{\pi} \cdot \left(\ln \boldsymbol{\pi} + \sum_{\tau=1}^t \mathbf{F}_\tau + \boldsymbol{\gamma} \cdot \mathbf{G} \right) + \mathcal{O}(\ln t)
\end{aligned}$$

where $F(\tau, \pi)$ and $G(\pi)$ are variational and expected free energies under a particular policy π , $\mathbf{F}_\tau = (F(\tau, \pi = 1), F(\tau, \pi = 2), \dots)^T$ and $\mathbf{G} = (G(\pi = 1), G(\pi = 2), \dots)^T$ are their vector representations, and $\boldsymbol{\gamma}$ is a precision. Thus, from $\partial F / \partial \boldsymbol{\pi} = 0$, the posterior belief about π is given by:

$$\boldsymbol{\pi} = \sigma \left(- \sum_{\tau=1}^t \mathbf{F}_\tau - \boldsymbol{\gamma} \cdot \mathbf{G} \right)$$

In this work, the policy is replaced with a sequence of decisions $\delta_1, \dots, \delta_t$, that minimise the risk associated with future outcomes. The generative model we consider can be related to the standard formulation as follows: the expected free energy comprises the sum of expected intrinsic (i.e., epistemic) and extrinsic (i.e., pragmatic) values, based on posterior expectations about the current state, \mathbf{s}_{t-1} . Crucially, expected state transitions depend on policies (i.e., decisions) that the agent entertains. Thus, the expected free energy is a function of \mathbf{s}_{t-1} and $\boldsymbol{\delta}_t$, which is approximated with leading order terms as follows:

$$G(\delta_t) = \mathbb{E}_Q[-\delta_t \cdot \ln C \mathbf{s}_{t-1}] = -\delta_t \cdot \ln \mathbf{C} \mathbf{s}_{t-1}$$

using the policy mapping \mathbf{C} . Additionally, this work averages the transition probabilities under $Q(\pi)$ to finesse computational complexity. Under this marginalisation, variational free energy is given by equation (14), or equivalently,

$$F(o_{1:t}, \boldsymbol{\vartheta}) = \sum_{\tau=1}^t F(\tau) + \sum_{\tau=1}^t \boldsymbol{\delta}_\tau \cdot (\ln \boldsymbol{\delta}_\tau - \boldsymbol{\gamma} \cdot \ln \mathbf{C} \mathbf{s}_{t-1}) + \mathcal{O}(\ln t)$$

Here, the precision $\boldsymbol{\gamma}$ corresponds to the risk $1 - 2\Gamma_t$. Similar to the derivation of $\boldsymbol{\pi}$, when $\boldsymbol{\gamma} = 1$, $\partial F / \partial \boldsymbol{\delta}_t = 0$ yields the posterior belief

$$\boldsymbol{\delta}_t = \sigma(\ln \mathbf{C} \mathbf{s}_{t-1})$$

as shown in equation (17). This shows that $\boldsymbol{\delta}_t$ is homologous to $\boldsymbol{\pi}$, where the expected free energy is implicit in $-\ln \mathbf{C} \mathbf{s}_{t-1}$.

Finally, the agent needs to learn the policy mapping \mathbf{C} . Although in usual treatments, G is computed using posterior expectations of likelihood and priors (\mathbf{A} and \mathbf{B}), here, the agent learns estimates \mathbf{C} directly, from $\partial F / \partial \mathbf{c} = 0$, as in equation (18). This postdiction approach simplifies the computation of G compared to evaluations based on \mathbf{A} and \mathbf{B} . From this perspective, the risk

can be viewed as negative precision. A lower risk (i.e., higher precision) facilitates more exploitive behaviour; conversely, a higher risk suppresses the current strategy—and induces a more explorative behaviour. In short, the proposed method—using postdiction of past experience as a proxy—simplifies the computational architecture for planning, rendering it more tractable for two-layer networks.”

Secondly, instead of explicit planning, this approach here appears to use a variational categorical distribution to directly compute actions via a method of postdiction of previous experiences. While this does not appear to be an unjustifiable method, a clear elucidation of the exact differences between the method here and more standard active inference would be much appreciated.

[Response]

In response to the above comment, we have tried to elucidate the correspondence between the current and standard active inference schemes (please see above). In short, the decision δ_t in this work is homologous to the policy π and the expected free energy is implicit in $-\ln \mathbf{C} \mathbf{s}_{t-1}$. This method using postdiction of previous experiences is a proxy for explicit planning in order to simplify the computational architecture to make it tractable for two-layer networks.

Additionally, this reviewer found the section concerning the computation of the action via the future risk and the variational policy hard to follow. I believe this would be much improved by additional intuition about why this counterfactual postdictive action selection mechanism is useful.

[Response]

Thank you for highlighting this. We have embellished the section on action selection by adding an intuitive explanation on the postdictive action selection. Please refer to the main text for details. For instance, we have added the following to Results and Methods sections:

“Equation (3) says that the probability of selecting decision δ_t after receiving s_{t-1} is determined by C when $\gamma_t = 0$, whereas it is inversely proportional to C (in the element-wise sense) when $\gamma_t = 1$.”

“Equation (3) represents a mechanism by which the agent learns to select the preferred decisions. This is achieved by updating the policy mapping C depending on the consequences of previous decisions.”

“Interestingly, this behavioural optimisation mechanism derived from the Bayesian inference turns out to be identical to a post hoc modulation of Hebbian plasticity (see equations (7) and (9), and Methods D).”

I also have some questions about the nature of the nonlinear terms in the dynamics. Specifically the inverse sigmoid 'leak terms'. These do not appear to be derivable via gradient descent on the free energy functional as provided which does not appear to possess any nonlinear terms indeed.

Where does this term arise from?

[Response]

Both the neural network cost function and variational free energy involve nonlinearities, with respect to the neural activity, by construction. The mechanism wherein the inverse sigmoid leak term is derived from the neural network cost function is described in Supplementary Methods 3, in response to the above comment. For variational free energy, the derivative of the Shannon entropy of the state posterior $-\mathbf{s}_\tau \cdot \ln \mathbf{s}_\tau$ yields the softmax nonlinearity (Friston, FitzGerald et al., Neural Comput, 2017), which turns out to correspond to the inverse sigmoid leak term because the hidden states take binary values in this setup. We have described this in Supplementary Methods 4 (please see below).

Also, the nonlinearity in the fixed point solution to the active inference and the canonical neural network equation are different -- the canonical neural network uses a sigmoid vs a softmax for active inference. Where do these differences come from? It seems difficult to imagine that the same cost function can give rise to these two different fixed points

[Response]

The fixed points of active inference and canonical neural networks are identical. To explain this clearly, we have added the following as Supplementary Methods and have cited it in the Methods section:

“4. Correspondence between fixed points of active inference and canonical neural networks

The fixed points of active inference and canonical neural networks are identical. The canonical neural network has the following fixed point (equation (19)):

$$x(t) = \text{sig}((W_1 - W_0)o(t) + (K_1 - K_0)x(t - \Delta t) + h_1 - h_0)$$

which can be expressed as

$$x(t) = \frac{\exp(W_1 o(t) + K_1 x(t - \Delta t) + h_1)}{\exp(W_1 o(t) + K_1 x(t - \Delta t) + h_1) + \exp(W_0 o(t) + K_0 x(t - \Delta t) + h_0)}$$

following the definition of the sigmoid function. The fixed point of active inference—that is, the posterior expectation of hidden states—is expressed using the softmax function as follows (equation (16)):

$$\mathbf{s}_t = \sigma(\ln \mathbf{A} \cdot o_t + \ln \mathbf{B}^\dagger \cdot \mathbf{s}_{t-1} + \ln D)$$

Because each element of the hidden states s_t takes a binary state (0 or 1), owing to the definition of the softmax function, we have

$$\begin{cases} \mathbf{s}_{t1} \propto \exp(\ln \mathbf{A}_1 \cdot o_t + \ln \mathbf{B}_1^\dagger \cdot \mathbf{s}_{t-1} + \ln D_1) \\ \mathbf{s}_{t0} \propto \exp(\ln \mathbf{A}_0 \cdot o_t + \ln \mathbf{B}_0^\dagger \cdot \mathbf{s}_{t-1} + \ln D_0) \end{cases}$$

where \mathbf{s}_{t1} and \mathbf{s}_{t0} are the posterior beliefs of s_t taking 1 or 0, respectively. Because $\mathbf{s}_{t1} + \mathbf{s}_{t0} = \vec{1}$ by construction, we obtain

$$\mathbf{s}_{t1} = \frac{\exp(\ln \mathbf{A}_1 \cdot o_t + \ln \mathbf{B}_1^\dagger \cdot \mathbf{s}_{t-1} + \ln D_1)}{\exp(\ln \mathbf{A}_1 \cdot o_t + \ln \mathbf{B}_1^\dagger \cdot \mathbf{s}_{t-1} + \ln D_1) + \exp(\ln \mathbf{A}_0 \cdot o_t + \ln \mathbf{B}_0^\dagger \cdot \mathbf{s}_{t-1} + \ln D_0)}$$

This expression discloses the equivalence between $x(t)$ and \mathbf{s}_{t1} . Namely, because $W_l o(t) + K_l x(t - \Delta t) + h_l = \ln \mathbf{A}_l \cdot o_t + \ln \mathbf{B}_l^\dagger \cdot \mathbf{s}_{t-1} + \ln D_l$ holds, $x(t)$ and \mathbf{s}_{t1} share the same functional form. **Table 1** summarises their formal correspondence. In summary, active inference and canonical neural networks share the same fixed points. Further details are provided in [17].”

Finally, while the authors are admirably clear about the assumptions their model requires (assumptions 1-3), I would have found very interesting a more detailed discussion of the nature and biological plausibility of these assumptions in the paper. Especially the idea that tall elements of the neural circuit minimize a single cost-function (assumption 1)

[Response]

Thank you for highlighting this interesting issue. We have added the following to the Methods section:

“We note that when neural activity and plasticity follow differential equations, without loss of generality, we can consider a common cost function for a sufficiently long training time t . When neural activity and plasticity are expressed as

$$\left\{ \begin{array}{l} \dot{x}(t) \propto f'(x(t), o(t), W) \\ \dot{W} \propto \frac{1}{t} \int_0^t g'(x(\tau), o(\tau), W) d\tau \end{array} \right. \quad (22)$$

a cost function of the following form is implied:

$$L = - \int_{t-\Delta t}^t f(x(\tau), o(\tau), W) d\tau - \int_0^{t-\Delta t} g(x(\tau), o(\tau), W) d\tau \quad (23)$$

From this cost function, one can derive gradient descent rules for both neural activity and plasticity in the limit of a large t . This owes to the different time scales of updating $x(t)$ and W , known as an adiabatic approximation. Therefore, for a sufficiently large t , the existence of cost functions that satisfy assumption 1 is a mathematical truism. It should be emphasised that unless f matches g , the neural and synaptic dynamics result in biased inference and learning; therefore, the cost function defined in equation (7) is apt.”

In general I found the paper relatively well written and straightforward to follow. I personally would have liked to see a little more intuition and motivation for the technical sections, especially

around active inference, and especially an expanded exploration of the differences in the action selection mechanism compared to standard active inference methods. I believe this paper makes an interesting and potentially influential contribution to the literature on active inference.

[Response]

Thank you for useful suggestions and for expressing the interest of our work. We hope that the above revisions are what you had in mind.

Other points

We have moved a footnote to the main text.

We have added the following paragraph to the Discussion to emphasise benefits of the proposed scheme:

“Crucially, the proposed equivalence guarantees that an arbitrary neural network that minimises its cost function—possibly implemented in neuromorphic hardware or biological organisms—can be cast as performing variational Bayesian inference. This notion can dramatically reduce the complexity of designing self-learning neuromorphic hardware to perform various types of tasks; therefore, it offers a simple architecture and low computational cost. This leads to a unified design principle for biologically-inspired machines such as neuromorphic hardware to perform statistically optimal inference, learning, prediction, planning, and decision making.”

References

31. Wieland, S., Schindler, S., Huber, C., Köhr, G., Oswald, M. J. & Kelsch, W. Phasic dopamine modifies sensory-driven output of striatal neurons through synaptic plasticity. *J. Neurosci.* **35**, 9946-9956 (2015).
32. Brzosko, Z., Zannone, S., Schultz, W., Clopath, C. & Paulsen, O. Sequential neuromodulation of Hebbian plasticity offers mechanism for effective reward-based navigation. *eLife* **6**, e27756 (2017).
36. Parr, T. & Friston, K. J. Uncertainty, epistemics and active inference. *J. R. Soc. Interface* **14**, 20170376 (2017).
45. Sutton, R. S. & Barto, A. G. *Reinforcement Learning*. MIT Press, Cambridge, MA, USA (1998).
46. Sajid, N., Ball, P. J., Parr, T. & Friston, K. J. Active inference: demystified and compared. *Neural Comput.* **33**, 674-712 (2021).
47. He, K. et al. Distinct eligibility traces for LTP and LTD in cortical synapses. *Neuron* **88**, 528-538 (2015).
52. Daunizeau, J., Den Ouden, H. E., Pessiglione, M., Kiebel, S. J., Stephan, K. E. & Friston, K. J. Observing the observer (I): Meta-Bayesian models of learning and decision-making. *PLoS One*

- 5, e15554 (2010).
53. Funamizu, A., Kuhn, B. & Doya, K. Neural substrate of dynamic Bayesian inference in the cerebral cortex. *Nat. Neurosci.* **19**, 1682-1689 (2016).
 60. Berkes, P., Orbán, G., Lengyel, M. & Fiser, J. Spontaneous cortical activity reveals hallmarks of an optimal internal model of the environment. *Science* **331**, 83-87 (2011).
 61. FitzHugh, R. Impulses and physiological states in theoretical models of nerve membrane. *Biophys. J.* **1**, 445-466 (1961).
 62. Nagumo, J., Arimoto, S. & Yoshizawa, S. An active pulse transmission line simulating nerve axon. *Proc. IRE* **50**, 2061-2070 (1962).
 63. Hindmarsh, J. L. & Rose, R. M. A model of neuronal bursting using three coupled first order differential equations. *Proc. R. Soc. Lond. B* **221**, 87-102 (1984).

Reviewers' comments:

Reviewer #1 (Remarks to the Author):

Isomura et al show that recurrent neural networks can perform active inference and learning in terms of the free energy minimization framework. This work closes an important gap in linking free energy minimization to the neural dynamics and circuit anatomy observed in biology. The paper is technically sound and well written. I'm left with only one comment to strengthen the context with prior existing literature.

The free energy minimization principle (FEP), that is used here, is very similar in nature to approximate Bayesian inference which has been extensively studied in computational neuroscience and therefore a large overlap between the proposed and earlier models exist. This prior work should be cited and discussed in the context of the new model to provide a broader overview of existing literature and their links to the FEP. Notably, similar neural network architectures for approximate Bayesian inference have been proposed earlier, e.g. Deneve et al. 2008., Brea et al. 2013, Rezende et al 2014., Kappel et al 2014. Brea et al. 2014 assumed variational approximations similar to the one used here. Kappel et al. 2014, demonstrated approximate Bayesian inference on exponential family distributions. Rückert et al. 2015 also demonstrated probabilistic planning with recurrent networks, similar to the FEP model presented here. References to these prior related work should be included and differences to the new model discussion.

References:

Brea, Johanni, Walter Senn, and Jean-Pascal Pfister. "Matching recall and storage in sequence learning with spiking neural networks." *Journal of neuroscience* 33.23 (2013): 9565-9575.

Deneve, Sophie. "Bayesian spiking neurons II: learning." *Neural computation* 20.1 (2008): 118-145.

D Kappel, B Nessler, W Maass. STDP installs in winner-take-all circuits an online approximation to hidden Markov model learning. *PLoS computational biology* 10 (3), e1003511

Jimenez Rezende, Danilo, and Wulfram Gerstner. "Stochastic variational learning in recurrent spiking networks." *Frontiers in computational neuroscience* 8 (2014): 38.

E Rueckert, D Kappel, D Tanneberg, D Pecevski, J Peters. Recurrent spiking networks solve planning tasks. *Scientific reports* 6 (1), 1-10

Reviewer #2 (Remarks to the Author):

In this paper, Isomura and colleagues prove a mathematical equivalence between a form of active inference and the dynamics (including plasticity) of a standard recurrent neural network. The paper contains multiple clever mathematical tricks and interesting insights spanning neural dynamics, plasticity and inference. In addition to the main result (which is this mathematical proof) the paper presents a neat proof-of-concept application with an agent solving a maze navigation task, and shows (intriguingly) that the framework can be used to predict performance of the agent in unseen mazes. Overall I think this is a good paper, and I think it should be published.

My main concern, echoing that of Reviewers #1 and #2, is that the paper is prohibitively dense and hard to follow for all readers except those that are already very familiar with the mathematics of active inference. The paper introduces a lot of mathematical notation, assumes a lot of background

knowledge, and the main results are very hard to grasp without such background knowledge (let alone the methods).

As another concern, it seems to me that assumption 2 (or rather, the implications of assumption 2, in Methods D) are quite problematic. In Methods D the authors assume that "the time constant of neural activity is smaller than that of sensory inputs," so that the updates to neural activity can be computed from fixed points. How valid is this assumption? On what basis is this assumed, and what happens if this assumption is not satisfied? We know neural activity is noisy and metastable, so it seems unrealistic to demand that neural activity is updated to the fixed point. The authors should at the very least add some comments on why this is justified and what happens if this assumption doesn't hold.

Finally, I thought the section on estimation of implicit priors was particularly interesting, yet also particularly opaque. This seems like a really valuable capability of the method, but both the results and corresponding methods sections are somewhat unclear, so I can't fully judge its potential. It would be very helpful if the authors could elaborate a bit more on this.

Other minor comments:

- The putative biological nature of the modulator γ , and its interpretation as an analogue of 'risk' in active inference should be explained in more detail.
- The definition of post_i in L. 376 contains $\{x(t), \bar{x}(t)\}$ twice, which makes sense but seems confusing at first. It would be helpful for readers if the authors could add a note explaining why this is written this way.
- In the discussion around Eq. (2) the matrix C is used (and described in passing as "policy mapping") much earlier than it is properly introduced. This is rather confusing, and I'd recommend introducing C properly before presenting the implementation of fictive causality.

Reviewer #3 (Remarks to the Author):

The authors explore the relationship between active inference and neural networks, proposing a duality between a specific type of POMDP and a recurrent NN. The ideas are interesting, but the paper is hard to follow for a number of reasons detailed below. In the following I put forward my main concerns about the paper, and then add a number of small comments.

Main concerns

1. My main concern is about how general the results are. The abstract suggests an equivalence between a "canonical" class of NN and POMDPs. However, the results themselves suggest a link between a very specific type of systems. Can one build a NN that is equivalent to any POMDPs, or vice versa? If not, what is the exact limit of the presented results. Unless this is explored, I believe the results are of interest for researchers that work with these actual type of models but not for people that don't. Also, some related questions:
 - Is the formulation of POMDP only valid for discrete variables?
 - Line 177 states that s_t , o_t , and δ_t are binary variables. Isn't this an important constraint too?
2. Related to the previous point, it would be important to clarify the relationship between the results presented in this paper and the complete class theorem. It seems that this theorem already guarantee

an equivalent between a broad class of NNs and variational Bayes. If so, what is the theoretical contribution of the presented results, beyond instantiating those results for a particular case? Related to this, it is not clear why this theorem does not suffice to establish a relationship between active inference and a given NN, which is nevertheless regarded as an open question in lines 70-71. Clarifying all these issues is important to understand the extend of the theoretical contribution.

3. I couldn't understand the maths of the paper due to a number of non-standard undefined terms. Please clarify the following:

- What is "Cat(A)"? Please define this. Also define what it is when it has a vector argument, as in line 216.

- When stating $P(o | s, A) = \text{Cat}(A)$, are you saying that A is a random variable and that the probability of o given s and A does not depend of o or s but only A?

- What is the "likelihood mapping"?

- In equation (3), is C' a scalar while C is a matrix? If so, please a more contrasting notation (maybe lowercase for C').

4. Despite some references in the discussion, the paper is focused on an equivalence between stochastic models, and has pretty indirect real relationship with biology. I believe it contents, as they stand now, would be of more interest to researchers working in AI and artificial life, rather than people working in biology. Consider including a biological case study, either including real data or a NN that has been fitted to real neuronal activity.

Minor observations

a) I found the ordering in which the results are presented very confusing. I'd suggest to first introduce the two models (the NN and the POMDP), and only then establish the equivalence. Otherwise, it is disorienting to have a statement that sounds very general and then present more specific models latter, which are actually constraining the scope of the results. Also, given that the narrative of the paper is to explain a NN in terms of active inference, it would make sense to first introduce the NN and then introduce the POMDP.

b) The retrodiction aspect of the POMDP felt as coming out of the blue. Please motivate why this is needed or desirable in this context.

c) In line 101, what does "c.f. assumption 1" mean? Is that pointing to an assumption that is somewhere else? Or establishing this as assumption 1? Please make this more clear.

d) It is not clear how assumption 1 necessarily implies gradient descend, as stated in lines 105-106.

e) In the caption of Fig.1 and other places, should it be "formation" or "formulation"?

f) Line 139: "the dynamical system" should be "a dynamical system"?

g) The first term in the product in equation (2) consider variables whose indices are 0. Please fix this.

19 September 2021

Takuya Isomura

Reviewers' comments:

Reviewer #1 (Remarks to the Author):

Isomura et al show that recurrent neural networks can perform active inference and learning in terms of the free energy minimization framework. This work closes an important gap in linking free energy minimization to the neural dynamics and circuit anatomy observed in biology. The paper is technically sound and well written. I'm left with only one comment to strengthen the context with prior existing literature.

[Response]

We would like to thank the reviewer for these gracious and valuable comments. As described below, we have revised the manuscript to address the issues raised. We hope these revisions are what you and the editor had in mind.

The free energy minimization principle (FEP), that is used here, is very similar in nature to approximate Bayesian inference which has been extensively studied in computational neuroscience and therefore a large overlap between the proposed and earlier models exist. This prior work should be cited and discussed in the context of the new model to provide a broader overview of existing literature and their links to the FEP. Notably, similar neural network architectures for approximate Bayesian inference have been proposed earlier, e.g. Deneve et al. 2008., Brea et al. 2013, Rezende et al 2014., Kappel et al 2014. Brea et al. 2014 assumed variational approximations similar to the one used here. Kappel et al. 2014, demonstrated approximate Bayesian inference on exponential family distributions. Rückert et al. 2015 also demonstrated probabilistic planning with recurrent networks, similar to the FEP model presented here. References to these prior related work should be included and differences to the new model discussion.

References:

Brea, Johanni, Walter Senn, and Jean-Pascal Pfister. "Matching recall and storage in sequence learning with spiking neural networks." *Journal of neuroscience* 33.23 (2013): 9565-9575.

Deneve, Sophie. "Bayesian spiking neurons II: learning." *Neural computation* 20.1 (2008): 118-145.

D Kappel, B Nessler, W Maass. STDP installs in winner-take-all circuits an online approximation to hidden Markov model learning. *PLoS computational biology* 10 (3), e1003511

Jimenez Rezende, Danilo, and Wulfram Gerstner. "Stochastic variational learning in recurrent spiking networks." *Frontiers in computational neuroscience* 8 (2014): 38.

E Rueckert, D Kappel, D Tanneberg, D Pecevski, J Peters. Recurrent spiking networks solve planning tasks. *Scientific reports* 6 (1), 1-10

[Response]

Thank you for this insightful comment. We have cited these references in the Introduction and Results sections as follows:

“This framework integrates perceptual (unsupervised), reward-based (reinforcement), and motor (supervised) learning in a unified formulation that shares many commitments with neuronal implementations of approximate Bayesian inference using spiking models [8–12].”

“In most formulations, active inference goes further than simply assuming action and perception minimise variational free energy—it also considers the consequences of action as minimising expected free energy, i.e., planning (and control) as inference, as a foundational approach to sentient behaviour [12,23–27].”

“In particular, we assume that $Q(\vartheta)$ is an exponential family (as considered in previous works [10]) and ...”

Additionally, we have discussed the difference between previous and present works in the Discussion section as follows:

“Based on the view of the brain as an agent that performs Bayesian inference, neuronal implementations of Bayesian belief updating have been proposed, which enables neural networks to store and recall spiking sequences [8], learn temporal dynamics and causal hierarchy [9], extract hidden causes [10], solve maze tasks [11], and make plans to control robots [12]. In these approaches, the update rules are generally derived from Bayesian cost functions (e.g., variational free energy). However, the precise relationship between these update rules and the neural activity and plasticity of canonical neural networks has yet to be fully established.”

“We identified a one-to-one correspondence between a neural network architecture and a specific POMDP implicit in that network. Equation (2) speaks to a unique POMDP model consistent with the neural network architecture defined in equation (6), where their correspondences are summarised in **Table 2**. This means that our scheme can be used to identify the form of POMDP, given an observable circuit structure. Moreover, the free parameters—that parameterise equation (6)—can be estimated using equation (24). This means that the generative model and ensuing

variational free energy can, in principle, be reconstructed from empirical data. This offers a formal characterisation of implicit Bayesian models entailed by neural circuits, thereby enabling a prediction of subsequent learning. Our numerical simulations, accompanied by previous work [22], show that canonical neural networks behave systematically—and in a distinct way—depending on the implicit POMDP (**Fig. 4**). This kind of self-organisation depends on implicit prior beliefs, which can therefore be characterised empirically (**Fig. 5**).”

Please also refer to our response to the Reviewer 3’s Comment #2, where we highlight the novelty of the present scheme.

Reviewer #2 (Remarks to the Author):

In this paper, Isomura and colleagues prove a mathematical equivalence between a form of active inference and the dynamics (including plasticity) of a standard recurrent neural network. The paper contains multiple clever mathematical tricks and interesting insights spanning neural dynamics, plasticity and inference. In addition to the main result (which is this mathematical proof) the paper presents a neat proof-of-concept application with an agent solving a maze navigation task, and shows (intriguingly) that the framework can be used to predict performance of the agent in unseen mazes. Overall I think this is a good paper, and I think it should be published.

[Response]

We would like to thank the reviewer for these gracious and valuable comments. As described below, we have revised the manuscript to address the issues raised. We hope these revisions are what you and the editor had in mind.

My main concern, echoing that of Reviewers #1 and #2, is that the paper is prohibitively dense and hard to follow for all readers except those that are already very familiar with the mathematics of active inference. The paper introduces a lot of mathematical notation, assumes a lot of background knowledge, and the main results are very hard to grasp without such background knowledge (let alone the methods).

[Response]

Thank you for pointing this out. To address this issue, we have summarised the mathematical notations in the newly created **Table 1**, which is now cited in the Results and Methods sections. In the table, we focus on explaining the mathematical notations highlighted by the reviewers (please

see our responses to these comments). We have included **Table 1** at the end of the manuscript.

Table 1. Glossary of expressions

Expression	Description
o_t, s_t, δ_t	Observations o_t , hidden states s_t , and decisions (actions) δ_t are random variables that follow categorical distributions. Each element of them takes 0 or 1.
Γ_t	Risk function Γ_t parameterises a categorical distribution over γ_t : $P(\gamma_t) = \text{Cat}\left(\left(\Gamma_t, \overline{\Gamma_t}\right)^T\right)$. This can be read as an arbitrary neuromodulator that regulates synaptic plasticity through equation (9), which becomes a risk function under the variational Bayes formulation.
A, B, C	Parameter matrices A , B , and C are random variables with Dirichlet distributions $P(A) = \text{Dir}(a)$, $P(B) = \text{Dir}(b)$, and $P(C) = \text{Dir}(c)$.
$\text{Cat}(A)$	Categorical distribution. In this expression, the probability that $o_\tau = i$ is realised given $s_\tau = j$ is A_{ij} ; that is, $P(o_\tau = i s_\tau = j, A) = A_{ij}$. For any pair of i and j , this is expressed as $P(o_\tau s_\tau, A) = \text{Cat}(A)$, where matrix A is the likelihood mapping that maps s_τ to o_τ . Note that $A_{k\vec{l}}^{(i)}$ means that the probability that $o_\tau^{(i)} = k$ is realised given $s_\tau = \vec{l} = (l_1, \dots, l_N)^T \in \{0,1\}^{N_s}$.

As another concern, it seems to me that assumption 2 (or rather, the implications of assumption 2, in Methods D) are quite problematic. In Methods D the authors assume that "the time constant of neural activity is smaller than that of sensory inputs," so that the updates to neural activity can be computed from fixed points. How valid is this assumption? On what basis is this assumed, and what happens if this assumption is not satisfied? We know neural activity is noisy and metastable, so it seems unrealistic to demand that neural activity is updated to the fixed point. The authors

should at the very least add some comments on why this is justified and what happens if this assumption doesn't hold.

[Response]

Thank you for pointing this out. First, we emphasise that the present equivalence is derived from the differential equation (equation (6)), but not from its fixed point (equation (19)); thus, the main claim of this paper holds true irrespective of the time constant of neurons.

As for the question “How valid is this assumption? On what basis is this assumed”, we have added the following to Methods D:

“In this section, we elaborate the one-to-one correspondences between components of canonical neural networks and variational Bayes, via an analytically tractable model. Neurons respond quickly to a continuous stimulus stream with a timescale faster than typical changes in sensory input. For instance, a peak of spiking responses in visual cortex (V1) appears between 50 and 80 ms after a visual stimulation [64,65], which is substantially faster than the temporal autocorrelation timescale of natural image sequences (approximately 500 ms) [66,67]. Thus, we consider the case where the neural activity converges to a fixed point, given a sensory stimulus.”

As for the question “what happens if this assumption is not satisfied?”, we have added the following to Methods D:

“We note that the present equivalence is derived from the differential equation (equation (6)), but not from its fixed point; thus, the equivalence holds true irrespective of the time constant of neurons. To rephrase, neural networks with a large time constant formally correspond to Bayesian belief updating with a large time constant, which implies an insufficient convergence of the posterior beliefs.”

As for the comment “neural activity is noisy and metastable, so it seems unrealistic to demand that neural activity is updated to the fixed point”, we note that the fixed-point equation (equation (19)) exhibits a dynamically changing neural activity upon receiving sensory stimuli and past neural activity through feedforward and recurrent synaptic connections. Thus, the equation indeed involves the dynamics driven by noisy sensory inputs and the metastability of states.

Additionally, we have revised the sentence as follows:

“When the neural activity asymptotically converges to a steady state, the fixed point of equation (6)—i.e., x and y that give $\dot{x} = 0$ and $\dot{y} = 0$ —is provided as follows:”

Finally, I thought the section on estimation of implicit priors was particularly interesting, yet also particularly opaque. This seems like a really valuable capability of the method, but both the results and corresponding methods sections are somewhat unclear, so I can't fully judge its potential. It would be very helpful if the authors could elaborate a bit more on this.

[Response]

Thank you for highlighting this issue. We appreciate that the reviewer found the present theory potentially variable for estimating implicit priors. To elaborate this further, we have created a separate Methods section “F. Protocols for data analyses” as follows:

“F. Protocols for data analyses

When the belief updating of implicit priors (D, E) is slow in relation to experimental observations, D and E (which are encoded by ϕ and ψ) can be viewed as being fixed over a short period of time, as an analogy to a homeostatic plasticity over longer time scales [68]. Thus, ϕ and ψ can be statistically estimated by a conventional Bayesian inference (or maximum likelihood estimation, given a flat prior) based on empirically observed neuronal responses. In this case, the estimators of ϕ and ψ are obtained as follows:

$$\begin{cases} \Phi = \ln\left(\frac{1}{t} \sum_{\tau=1}^t \left(\frac{x_t}{x_t}\right)\right) \\ \Psi = \ln\left(\frac{1}{t} \sum_{\tau=1}^t \left(\frac{y_t}{y_t}\right)\right) \end{cases} \quad (24)$$

Note that we suppose constraints $\exp(\phi_1) + \exp(\phi_0) = \vec{1}$ and $\exp(\psi_1) + \exp(\psi_0) = \vec{1}$. This assumption finesses the estimation of implicit priors (**Fig. 5a**)—owing to the equivalence $\phi = \ln D$ and $\psi = \ln E$ (see **Fig. 3** and **Table 2**)—and the ensuing reconstruction of variational free energy, which was conducted by substituting equation (24) into equation (15). Equation (24) indicates that the average activity encodes prior beliefs, which is consistent with the experimental observations; in that the activity of sensory areas reflects prior beliefs [69].

Furthermore, given a generative model, the posterior beliefs of external states were updated based on a sequence of sensory inputs by minimising variational free energy, without reference to neural activity or behaviour. Thus, the reconstructed variational free energy enabled us to predict subsequent inference and learning of the agent, without observing neural activity or behaviour (**Fig. 5b**). In **Fig. 5**, for simplicity, the form of the risk function was supposed to be known when reconstructing the cost function. Although we did not estimate ϕ in **Fig. 5**, the previous work showed that our approach can estimate ϕ from simulated neural activity data [22].”

Other minor comments:

- The putative biological nature of the modulator γ , and its interpretation as an analogue of 'risk' in active inference should be explained in more detail.

[Response]

Thank you. We have added the following revised sentences and tables to describe their relationships in more detail:

“The agent makes decisions to minimise a risk function $\Gamma_t := \Gamma(o_{1:t}, \mathbf{s}_{1:t}, \boldsymbol{\delta}_{1:t-1}, \boldsymbol{\theta})$ that it employs (where $0 \leq \Gamma_t \leq 1$; see **Table 1**).”

“This means that when $(x(\tau)^T, \bar{x}(\tau)^T)^T = \mathbf{s}_\tau$, $(y(\tau)^T, \bar{y}(\tau)^T)^T = \boldsymbol{\delta}_\tau$, $\Gamma(t) = \Gamma_t$, $\widehat{W}_l = \mathbf{A}_{1l}^T$, $\widehat{K}_l = \mathbf{B}_{1l}^{\dagger T}$, and $\widehat{V}_l = \mathbf{C}_{1l}^{\dagger T}$ (for $l = 0,1$), the neural network cost function is identical to variational free energy, ...”

“Further, an arbitrary neuromodulator that regulates synaptic plasticity as depicted in equation (9) plays the role of the risk function in the POMDP model defined in equation (2), where the equivalence can be confirmed by comparing equations (9) and (18).”

Table 1. Glossary of expressions

Expression	Description
Γ_t	Risk function Γ_t parameterises a categorical distribution over γ_t : $P(\gamma_t) = \text{Cat}\left(\left(\Gamma_t, \overline{\Gamma}_t\right)^T\right)$. This can be read as an arbitrary neuromodulator that regulates synaptic plasticity through equation (9), which becomes a risk function under the variational Bayes formulation.

Table 2. Correspondence of variables and functions

Neural network formation	Variational Bayes formation
Neuromodulator	$\Gamma(t) \Leftrightarrow \Gamma_t$ Risk function

- The definition of post_i in L. 376 contains $\{x(t), \bar{x}(t)\}$ twice, which makes sense but seems confusing at first. It would be helpful for readers if the authors could add a note explaining why this is written this way.

[Response]

Thank you. We have added the following in Results section:

“Note that some variables (e.g., $x(t)$) appear several times because some synapses connect to the same pre- or post-synaptic neurons as other synapses.”

- In the discussion around Eq. (2) the matrix C is used (and described in passing as "policy mapping") much earlier than it is properly introduced. This is rather confusing, and I'd recommend introducing C properly before presenting the implementation of fictive causality.

[Response]

Thank you for your comment. To introduce C properly before equation (2), we have added the following:

“Moreover, decision at time τ is conditioned on the previous state $s_{\tau-1}$ and current risk γ_t , characterised by the policy mapping C , $P(\delta_\tau | s_{\tau-1}, \gamma_t, C)$, where γ_t contextualises $P(\delta_\tau | s_{\tau-1}, \gamma_t, C)$ as described below.”

Reviewer #3 (Remarks to the Author):

The authors explore the relationship between active inference and neural networks, proposing a duality between a specific type of POMDP and a recurrent NN. The ideas are interesting, but the paper is hard to follow for a number of reasons detailed below. In the following I put forward my main concerns about the paper, and then add a number of small comments.

[Response]

We would like to thank the reviewer for their effort in reviewing and commenting on our manuscript. As described below, we have revised the manuscript to address the issues raised. We hope these revisions are what you and the editor had in mind.

Main concerns

1. My main concern is about how general the results are. The abstract suggests an equivalence between a “canonical” class of NN and POMDPs. However, the results themselves suggest a link between a very specific type of systems. Can one build a NN that is equivalent to any POMDPs, or vice versa? If not, what is the exact limit of the presented results. Unless this is explored, I believe the results are of interest for researchers that work with these actual type of models but not for people that don't. Also, some related questions:

[Response]

Thank you for your comment. In the first sub-section in the Results section, we show that any neural network that minimises a cost function can be cast as performing variational Bayesian inference. This owes to the complete class theorem [14–16]. Thus, the condition for this proposition is that a neural network minimises a cost function (which we referred to as assumption 1). This condition is explicitly written in the Abstract as “This work considers a class of canonical neural networks comprising rate coding models, wherein neural activity and plasticity minimise a common cost function—and plasticity is modulated with a certain delay.” Please refer to Supplementary Methods 1 for technical details.

Subsequently, as the main result, we show that a class of canonical recurrent networks with a sigmoidal activation function can be cast as variational Bayesian inference under a particular form of the POMDP generative model, irrespective of the details of its network structures and parameters. Please note that this class of recurrent neural networks covers any hierarchical network structures because such structures can be expressed by simply setting some synaptic strengths to zero; thus, this is a fairly generic model.

As for the question “Can one build a NN that is equivalent to any POMDPs, or vice versa?”, there exists a generative model (which may not be limited to conventional POMDP models) that formally corresponds to any neural network that minimises a cost function according to the complete class theorem, as we described in the Results section. Conversely, whether a neural network can implement any POMDP is outside the scope of this work.

As for the comment “Unless this is explored, I believe the results are of interest for researchers that work with these actual type of models but not for people that don’t”, we have conducted new analyses and have shown that the neural network model considered in this work is derived from the Hodgkin-Huxley model and the FitzHugh-Nagumo model through some approximations. This was appended as Supplementary Methods 3; please refer to our response to Comment #4 for details. Hence, we believe that this is sufficiently canonical and concise expression of a network of biologically plausible neuron models, which can appeal to a broad audience in neurobiology.

- Is the formulation of POMDP only valid for discrete variables?

[Response]

It is true that a POMDP assumes a discrete state space; however, this does not mean that the present equivalence holds true only in discrete environments. This work shows a formal correspondence between a neural network architecture and a generative model (i.e., a hypothesis about the external milieu system) that the agent employs. It is noteworthy that a generative model is not necessarily supposed to match the actual generative process of the external milieu.

Therefore, the present equivalence is valid irrespective of the actual external milieu system. Further descriptions are provided in the previous paper [22].

- Line 177 states that s_t , o_t , and δ_t are binary variables. Isn't this an important constraint too?

[Response]

This constraint serves to specify a class of POMDP models that formally correspond to a class of canonical neural networks with a sigmoid activation function. To rephrase, a class of canonical neural networks that we consider in this work is fully and sufficiently explained by a POMDP model with binary state variables, meaning that this characterisation does not limit the application range of the present theory. We have emphasised this in the Methods section as follows:

“As described in the Results section, this form of generative model is suitable to characterise a class of canonical neural networks defined by equation (6). This means that none of the aforementioned assumptions—regarding the generative model—limit the scope of the proposed equivalence between neural networks and variational Bayesian inference, as long as neural networks satisfy assumptions 1–3.”

2. Related to the previous point, it would be important to clarify the relationship between the results presented in this paper and the complete class theorem. It seems that this theorem already guarantee an equivalent between a broad class of NNs and variational Bayes. If so, what is the theoretical contribution of the presented results, beyond instantiating those results for a particular case? Related to this, it is not clear why this theorem does not suffice to establish a relationship between active inference and a given NN, which is nevertheless regarded as an open question in lines 70-71. Clarifying all these issues is important to understand the extend of the theoretical contribution.

[Response]

Thank you for highlighting this important issue. As for the questions “what is the theoretical contribution of the presented results” and “why this theorem does not suffice to establish a relationship between active inference and a given NN”, the complete class theorem does not specify a form of generative model. Thus, the identification of a generative model that corresponds to a canonical neural network is a more difficult problem—one that we have addressed in this work. To address the question, we have further emphasised this in the Discussion section as follows:

“The complete class theorem [19–21] ensures that any neural network, whose activity and

plasticity minimise the same cost function, can be cast as performing Bayesian inference. However, identifying the implicit generative model that underwrites any canonical neural network is a more delicate problem because the theorem does not specify a form of a generative model for a given canonical neural network. The posterior beliefs are largely shaped by prior beliefs, making it challenging to identify the generative model by simply observing systemic dynamics. To this end, it is necessary to commit to a particular form of the generative model and elucidate how the posterior beliefs are encoded or parameterised by the neural network states. This work addresses these issues by establishing a reverse-engineering approach to identify a generative model implicit in a canonical neural network, thereby establishing one-to-one correspondences between their components. Remarkably, a network of rate coding models with a sigmoid activation function formally corresponds to a class of POMDP models, which provide an analytically tractable example of the present equivalence (please refer to the previous paper [22] for further discussion)."

3. I couldn't understand the maths of the paper due to a number of non-standard undefined terms. Please clarify the following:

- What is "Cat(A)"? Please define this. Also define what it is when it has a vector argument, as in line 216.
- When stating $P(o | s, A) = \text{Cat}(A)$, are you saying that A is a random variable and that the probability of o given s and A does not depend of o or s but only A ?
- What is the "likelihood mapping"?

[Response]

Thank you for pointing this out. We have summarised the mathematical expressions in the newly created **Table 1** in response to Reviewer 2's comments. As for the definitions of A and $\text{Cat}(A)$, we have added the following description in **Table 1** and cited it in the Results and Methods sections:

"The generation of observations $o_\tau := (o_\tau^{(1)}, \dots, o_\tau^{(N_o)})^T$ from external or hidden states milieu $s_\tau := (s_\tau^{(1)}, \dots, s_\tau^{(N_s)})^T$ is expressed in the form of a categorical distribution, $P(o_\tau | s_\tau, A) = \text{Cat}(A)$, where matrix A is also known as the likelihood mapping (see also **Table 1**). Here, A_{ij} means the probability that $o_\tau = i$ is realised given $s_\tau = j, \dots$ "

" $P(\theta)$ and $P(\gamma_t)$ are the prior distributions of the parameters and the risk (see **Table 1**)."

"The probability of an observation is determined by the likelihood mapping, from s_t to $o_t^{(i)}$, in terms of a categorical distribution: $P(o_t^{(i)} | s_t, A^{(i)}) = \text{Cat}(A^{(i)})$, where the elements of $A^{(i)}$ are

given by $A_{1\vec{l}}^{(i)} = P(o_t^{(i)} = 1 | s_t = \vec{l}, A^{(i)})$ and $A_{0\vec{l}}^{(i)} = 1 - A_{1\vec{l}}^{(i)}$ (see also **Table 1**). This encodes the probability that $o_t^{(i)}$ takes 1 or 0 when $s_t = \vec{l} = (l_1, \dots, l_N)^T \in \{0,1\}^{N_s}$.

Table 1. Glossary of expressions

Expression	Description
A, B, C	Parameter matrices A , B , and C are random variables with Dirichlet distributions $P(A) = \text{Dir}(a)$, $P(B) = \text{Dir}(b)$, and $P(C) = \text{Dir}(c)$.
$\text{Cat}(A)$	Categorical distribution. In this expression, the probability that $o_\tau = i$ is realised given $s_\tau = j$ is A_{ij} ; that is, $P(o_\tau = i s_\tau = j, A) = A_{ij}$. For any pair of i and j , this is expressed as $P(o_\tau s_\tau, A) = \text{Cat}(A)$, where matrix A is the likelihood mapping that maps s_τ to o_τ . Note that $A_{k\vec{l}}^{(i)}$ means that the probability that $o_\tau^{(i)} = k$ is realised given $s_\tau = \vec{l} = (l_1, \dots, l_N)^T \in \{0,1\}^{N_s}$.

- In equation (3), is C' a scalar while C is a matrix? If so, please a more contrasting notation (maybe lowercase for C').

[Response]

Both C and C' are matrices. We have emphasised this in the Results section as follows:

“We note that matrix C' denotes a normalisation factor that can be dropped from the following formulations, ...”

4. Despite some references in the discussion, the paper is focused on an equivalence between stochastic models, and has pretty indirect real relationship with biology. I believe it contents, as they stand now, would be of more interest to researchers working in AI and artificial life, rather than people working in biology. Consider including a biological case study, either including real data or a NN that has been fitted to real neuronal activity.

[Response]

Although we appreciate this comment, the equivalence we present is not between stochastic

models. We show a mathematical equivalence between variational Bayesian inference under a POMDP model (which is a stochastic model) and a neural network with an inverse sigmoid leak current, which is derived as a reduction of a realistic neuron model. This speaks to the biological plausibility of the considered model. To emphasise this further, and to address the reviewer’s concerns, we have conducted analyses on realistic models of neuronal activity. We have newly created Supplementary Methods 3—where we explicitly show that the canonical neural network considered in this work is derived from the Hodgkin-Huxley model and the FitzHugh-Nagumo model through some approximations—and cited it in the Results section as follows:

“This model is derived as a reduction of a realistic neuron model through some approximations (see Supplementary Methods 3 for details).”

“3. Derivation of canonical neural network

The Hodgkin-Huxley model comprises the following four differential equations that express the dynamics of membrane potential (v) and sodium (m, h) and potassium (n) ion channels [70]:

$$\begin{cases} C_M \dot{v} = -g_{Na} m^3 h (v - E_{Na}) - g_K n^4 (v - E_K) - g_L (v - E_L) + I \\ \dot{m} = \alpha_m(v)(1 - m) - \beta_m(v)m \\ \dot{h} = \alpha_h(v)(1 - h) - \beta_h(v)h \\ \dot{n} = \alpha_n(v)(1 - n) - \beta_n(v)n \end{cases} \quad (31)$$

Here, m, h, n denote the gating variables, unlike the notation in the main text; C_M is a membrane capacitance; I is the external input current; $\{E_{Na}, E_K, E_L\}$ and $\{g_{Na}, g_K, g_L\}$ are the reversal potential and conductance for sodium, potassium, and leak channels, respectively; and $\{\alpha_m(v), \alpha_h(v), \alpha_n(v), \beta_m(v), \beta_h(v), \beta_n(v)\}$ are the functions of v that characterise the dynamics of gating variables. Because m is known to exhibit a sufficiently faster dynamics compared to other variables, it can be treated as an instantaneous variable $m =$

$(1 + \beta_m(v)/\alpha_m(v))^{-1}$ (a quasi-steady state approximation) [71]. Further, n and $1 - h$ are

known to have similar shapes and time constants, $n \approx 1 - h$. Thus, by introducing a new effective variable u that interpolates between n and $1 - h$ (e.g., $u := (n + 1 - h)/2$), the original four equations are reduced to the following two equations (the two-dimensional Hodgkin-Huxley model) [71]:

$$\begin{cases} \dot{v} \propto f_v(v, u) + I \\ \dot{u} \propto f_u(v, u) \end{cases} \quad (32)$$

where new functions $f_v(v, u)$ and $f_u(v, u)$ characterise the effective dynamics of \dot{v} and $\dot{n} \approx -\dot{h}$, respectively. The nullclines (i.e., ensembles of points with $\dot{v} = 0$ and $\dot{u} = 0$) of equation (32) are depicted in **Fig. S1a**.

Supplementary Fig. S1. Comparisons of neuron models. Panels show the two-dimensional Hodgkin-Huxley model (a), FitzHugh-Nagumo model (b), and canonical neuron model considered in this work (c). In each panel, the v -nullcline (solid line) is approximately a cubic function of v , whereas the u -nullcline (dashed line) is approximately proportional to v . The colour map and arrows represent the velocity map in the (v, u) plane.

The FitzHugh-Nagumo model is a famous example of two-dimensional neuron models [72,73], which corresponds to the case where $f_v(v, u)$ and $f_u(v, u)$ in equation (32) are given as

$$\begin{cases} f_v(v, u) = v - \frac{v^3}{3} - u \\ f_u(v, u) = \alpha + \beta v - u \end{cases} \quad (33)$$

with parameters α and β (up to the scale of v). This model can be viewed as an approximation of the Hodgkin-Huxley model. The nullclines of the FitzHugh-Nagumo model (**Fig. S1b**) exhibit fairly similar shapes to those of the Hodgkin-Huxley model (**Fig. S1a**), meaning that the former well captures qualitative dynamical properties of the latter.

The canonical neural network with the inverse sigmoid leak current that we consider in this work is a family of the aforementioned models. Because the Taylor expansion provides a cubic approximation of the inverse sigmoid (or logit) function $\text{sig}^{-1}(x) = 4\left(x - \frac{1}{2}\right) + \frac{16}{3}\left(x - \frac{1}{2}\right)^3 + \mathcal{O}\left(\left(x - \frac{1}{2}\right)^5\right)$, one can characterise $f_v(v, u)$ using the inverse sigmoid function, instead of v^3 .

The Taylor expansion also provides $u(t + \Delta t) = u + \dot{u}\Delta t + \mathcal{O}(\Delta t^2)$; thus, one can replace u 's update rule $\dot{u} \propto f_u(v, u) = \alpha + \beta v - u$ with $u(t + \Delta t) = \alpha + \beta v$, or equivalently, $u = \alpha + \beta v(t - \Delta t)$, for a small Δt . Similarly, because $v(t - \Delta t) = v - \dot{v}\Delta t + \mathcal{O}(\Delta t^2)$, the linear v term in $f_v(v, u)$ can be replaced with $v(t - \Delta t)$. Hence, the FitzHugh-Nagumo model is approximated as follows (called as canonical neuron model here):

$$\begin{cases} \dot{v} \propto -\frac{v_{\text{PP}}^3}{16} \text{sig}^{-1}\left(\frac{v}{v_{\text{PP}}} + \frac{1}{2}\right) + \left(\frac{v_{\text{PP}}^2}{4} + 1\right) v(t - \Delta t) - u + I \\ u = \alpha + \beta v(t - \Delta t) \end{cases} \quad (34)$$

where positive constant v_{PP} denotes the peak-to-peak value of v . This is a delayed differential equation of a single variable v . The nullclines (i.e., ensembles of points with $\dot{v} = 0$ and $v = v(t - \Delta t)$) of equation (34) (**Fig. S1c**) precisely match those of the FitzHugh-Nagumo model (**Fig. S1b**), and they are fairly similar to those of the Hodgkin-Huxley model (**Fig. S1a**). This speaks to the biological plausibility of equation (34).

Finally, we introduce a new effective variable (firing intensity) $x := \frac{v}{v_{\text{PP}}} + \frac{1}{2}$, an effective auto-excitation coefficient $\kappa := \frac{16}{v_{\text{PP}}^2} \left(\frac{v_{\text{PP}}^2}{4} + 1 - \beta\right)$, and a new external input $I' := \frac{16}{v_{\text{PP}}^3} \left\{ I - \frac{1}{2} \left(\frac{v_{\text{PP}}^2}{4} + 1 - \beta\right) - \alpha \right\}$. Plugging u 's equation into v 's equation, equation (34) provides the canonical neuron model:

$$\dot{x} \propto -\text{sig}^{-1}(x) + \kappa x(t - \Delta t) + I' \quad (35)$$

A network of this model provides equation (6) in the main text, which we refer to as the canonical neural network, where I' represents the sum of synaptic inputs from feedforward and recurrent connections and firing thresholds. In summary, the canonical neural network—considered in this work—can be derived from the Hodgkin-Huxley model and the FitzHugh-Nagumo model through some approximations. In short, they comprise a family of biologically plausible neuron models.”

Minor observations

a) I found the ordering in which the results are presented very confusing. I'd suggest to first introduce the two models (the NN and the POMDP), and only then establish the equivalence. Otherwise, it is disorienting to have a statement that sounds very general and then present more specific models latter, which are actually constraining the scope of the results. Also, given that the narrative of the paper is to explain a NN in terms of active inference, it would make sense to first introduce the NN and then introduce the POMDP.

[Response]

Because of the property of the complete class theorem described above in response to Comment #2, we first established a general equivalence between a class of neural networks that minimise a cost function and variational Bayesian inference (which does not specify a form of generative model). Then, we describe the equivalence between canonical neural networks with a

sigmoid activation function and variational Bayesian inference under a POMDP generative model by explicitly showing one-to-one correspondences between their components. These are two different levels of conceptualisation of the present theory, which does not limit the scope of the results.

Additionally, to specify an external milieu system, we first need to introduce a POMDP model. Thus, it is natural to introduce variational Bayesian inference of the external milieu system immediately after that because their notations largely overlap. Subsequently, we can properly introduce canonical neural networks and the ensuing neural network cost function because, unless variational free energy is introduced, the motivation to consider a cost function for neural networks becomes unclear.

In summary, because introducing an external system and explaining its variational Bayesian inference is tightly connected, they should be described together immediately after establishing a general equivalence. Hence, we believe that the current order of presentation “general equivalence \rightarrow a POMDP model \rightarrow variational Bayesian inference \rightarrow neural network and its cost function \rightarrow specific equivalence” minimises the back and forth of explanations, therefore suitable to explain this work. Please note that, as for the general equivalence, we started with the neural network and then introduced variational inference as the reviewer suggests.

b) The retrodiction aspect of the POMDP felt as coming out of the blue. Please motivate why this is needed or desirable in this context.

[Response]

Thank you for pointing this out. In the Results section, we have emphasised the motivation as follows:

“To characterise optimal decisions as minimising expected risk, in our POMDP model, we use a fictive mapping from the current risk Γ_t to past decisions $\delta_1, \dots, \delta_{t-1}$ (**Fig. 2**).”

“We select this form of generative model because it speaks to the neuromodulation of synaptic plasticity, as shown in the next section.”

c) In line 101, what does “c.f. assumption 1” mean? Is that pointing to an assumption that is somewhere else? Or establishing this as assumption 1? Please make this more clear.

[Response]

We have revised the concerned phrase as follows:

“(c.f., assumption 1 specified in Introduction)”.

d) It is not clear how assumption 1 necessarily implies gradient descent, as stated in lines 105-106.

[Response]

Gradient descent is a generic local optimisation scheme to minimise a cost function. We suspect that the reviewer's question is whether there is a possibility that neural networks employ a minimisation scheme other than gradient descent (or related methods that use gradients), which means a non-local optimisation scheme. However, because the internal states of neural networks—including neural activity and synaptic strengths—exhibit continuous dynamics, cost function minimisation through non-local optimisation is not physiologically plausible. We have revised the sentence as follows:

“Based on assumption 1 and continuous updating nature of φ , the update rule for the i -th component of φ is derived as the gradient descent on the cost function, $\dot{\varphi}_i \propto -\partial L/\partial \varphi_i$.”

e) In the caption of Fig.1 and other places, should it be “formation” or “formulation”?

[Response]

The schematics depicted in Fig. 1 are not the formula. As shown in the forms of the neural network and Bayesian model of Fig. 1, we used ‘formation’ instead of ‘formulation’ to ensure clarity.

f) Line 139: “the dynamical system” should be “a dynamical system”?

[Response]

Thank you. We have fixed the typo.

g) The first term in the product in equation (2) consider variables whose indices are 0. Please fix this.

[Response]

Thank you for pointing this out. We have added the following after equation (2) to specify the treatment of conditional probabilities with variables having zero index:

“... and $P(s_1|s_0, \delta_0, B) = P(s_1)$ and $P(\delta_1|s_0, \gamma_t, C) = P(\delta_1)$ denote probabilities at $\tau = 1$.”

References

8. Brea, J., Senn, W. & Pfister, J. P. Matching recall and storage in sequence learning with spiking neural networks. *J. Neurosci.* **33**(23), 9565-9575 (2013).
9. Deneve, S. Bayesian spiking neurons II: learning. *Neural Comput.* **20**(1), 118-145 (2008).
10. Kappel, D., Nessler, B. & Maass, W. STDP installs in winner-take-all circuits an online approximation to hidden Markov model learning. *PLoS Comput. Biol.* **10**(3), e1003511 (2014).
11. Jimenez Rezende, D. & Gerstner, W. Stochastic variational learning in recurrent spiking networks. *Front. Comput. Neurosci.* **8**, 38 (2014).
12. Rueckert, E., Kappel, D., Tanneberg, D., Pecevski, D. & Peters, J. Recurrent spiking networks solve planning tasks. *Sci. Rep.* **6**(1), 1-10 (2016).
64. Lamme, V. A. & Roelfsema, P. R. The distinct modes of vision offered by feedforward and recurrent processing. *Trends Neurosci.* **23**(11), 571-579 (2000).
65. Benucci, A., Ringach, D. L. & Carandini, M. Coding of stimulus sequences by population responses in visual cortex. *Nat. Neurosci.* **12**(10), 1317-1324 (2009).
66. David, S. V., Vinje, W. E. & Gallant, J. L. Natural stimulus statistics alter the receptive field structure of v1 neurons. *J. Neurosci.* **24**(31), 6991-7006 (2004).
67. Bull, D. *Communicating pictures: A course in Image and Video Coding*. Academic Press (2014).
70. Hodgkin, A. L. & Huxley, A. F. A quantitative description of membrane current and its application to conduction and excitation in nerve. *J. Physiol.* **117**(4), 500-544 (1952).
71. Gerstner, W., Kistler, W. M., Naud, R. & Paninski, L. *Neuronal Dynamics: From Single Neurons to Networks and Models of Cognition*. Cambridge University Press (2014).
72. FitzHugh, R. Impulses and physiological states in theoretical models of nerve membrane. *Biophys. J.* **1**, 445-466 (1961).
73. Nagumo, J., Arimoto, S. & Yoshizawa, S. An active pulse transmission line simulating nerve axon. *Proc. IRE* **50**, 2061-2070 (1962).

Reviewers' comments:

Reviewer #1 (Remarks to the Author):

All my concerns have been addressed. I have no further comments.

Reviewer #3 (Remarks to the Author):

I appreciate the work done by the authors addressing my comments. In particular, I believe the additional developments included to this version are helpful for the paper. However, there are a few of my comments that have not been fully clarified. To be clear, I believe the paper provides valuable results and should be published, but I believe that there are still some issues that should be addressed before the paper is in good shape for readers. Below I state my outstanding concerns.

1. The goal of the paper is still not clearly stated in the introduction. In particular, The paragraph that starts in line 60 seems to be outlining previous work of the authors, but this is not 100% clear in the subsequent lines; please clarify this. Then, the paragraph that starts in line 74 states a side goal of the paper (demonstrating the ability of NNs to plan and minimise future risk) before the main goal of the paper has been stated. Then, the paragraph in line 87 describes what the paper does, but not what is the purpose behind all this. In response to my question about this, authors have added a clarification in the discussion. I believe this is insufficient, as it is crucial for the introduction to clearly state the goal of the paper — and how it relates to previous work. In particular, one confusing fact is that the title suggests the paper main achievement is demonstrating that NNs perform active inference, but (unless I'm mistaken) this seems to be a corollary of the complete class theorem. I'm sorry to bother with this, but I fear that if I'm confused with this, many readers will be as well.

2. The term "canonical neural network" has an important role in the paper, but I don't see what is about them that deserves the term "canonical". My belief is that most readers will believe "canonical NN" is a deep CNN or something of that sort. I'd suggest the authors to use another terminology, or to clarify their choice.

3. I still believe the term "likelihood mapping" should be clarified. Additionally, the expression $p(o_{\tau} | s_{\tau}, A)$ suggests that A is a random variable, but it seems A is just a matrix of transition probabilities. Also, note that $p(o_{\tau} | s_{\tau}, A)$ is a conditional probability which changes for each value of s_{τ} , hence I believe the notation the notation $\text{cat}(A)$ suggest that it doesn't depend on s_{τ} but only in A (if it would be a random variable. One potential solution could be to use the notation $p(o_{\tau} | s_{\tau}; A)$, with the ";" separating random variables from parameters. Another possible solution would be to use A as a subscript of p . Beyond the specific solution, it has to be made clear how to differentiate between random variables and parameters. Additionally, from the table, what does it mean for a probability to become "realised"?

8 November 2021

Takuya Isomura

Reviewers' comments:

Reviewer #1 (Remarks to the Author):

All my concerns have been addressed. I have no further comments.

[Response]

Thank you very much for confirming that our revisions are satisfactory. Your comments have been extremely helpful.

Reviewer #3 (Remarks to the Author):

I appreciate the work done by the authors addressing my comments. In particular, I believe the additional developments included to this version are helpful for the paper. However, there are a few of my comments that have not been fully clarified. To be clear, I believe the paper provides valuable results and should be published, but I believe that there are still some issues that should be addressed before the paper is in good shape for readers. Below I state my outstanding concerns.

[Response]

We would like to thank the reviewer for their additional and useful comments. As described below, we have revised the manuscript to address the issues raised. We hope that these revisions meet your expectations.

1. The goal of the paper is still not clearly stated in the introduction. In particular, The paragraph that starts in line 60 seems to be outlining previous work of the authors, but this is not 100% clear in the subsequent lines; please clarify this.

[Response]

Thank you for this comment. The paragraph from line 60 describes the previous work. To clarify this, we have modified the paragraph as follows:

“Crucially, as a corollary of the complete class theorem [19–21], any neural network minimising a cost function can be viewed as performing variational Bayesian inference, under some prior beliefs. We have previously introduced a reverse engineering approach that identifies a class of

biologically plausible cost functions for neural networks [22]. This foundational work identified a class of cost functions for single-layer feedforward neural networks of rate coding models with a sigmoid (or logistic) activation function—based on the assumption that the dynamics of neurons and synapses follow a gradient descent on a common cost function. We subsequently demonstrated the mathematical equivalence between the class of cost functions for such neural networks and variational free energy under a particular form of generative model. This equivalence licenses variational Bayesian inference as a fundamental optimisation process that underlies both the dynamics and function of such neural networks. Moreover, it enables one to characterise any variables and constants in the network in terms of quantities (e.g., priors) that underwrite variational Bayesian inference [22]. However, it remains to be established whether active inference is an apt explanation for any given neural network that actively exchanges with its environment. In this paper, we address this enactive or control aspect to complete the formal equivalence of neural network optimisation and the free-energy principle.”

Then, the paragraph that starts in line 74 states a side goal of the paper (demonstrating the ability of NNs to plan and minimise future risk) before the main goal of the paper has been stated. Then, the paragraph in line 87 describes what the paper does, but not what is the purpose behind all this. In response to my question about this, authors have added a clarification in the discussion. I believe this is insufficient, as it is crucial for the introduction to clearly state the goal of the paper — and how it relates to previous work.

[Response]

Thank you. To show that neural networks perform active inference, one needs to address planning. Thus, the point described in the paragraph from line 74 is essential for our main claim. This paragraph describes the background and identifies the research gap that we have addressed. To clarify this, we have modified the paragraphs as follows:

“In most formulations, active inference goes further than simply assuming action and perception minimise variational free energy—it also considers the consequences of action as minimising expected free energy, i.e., planning (and control) as inference, as a foundational approach to sentient behaviour [12,23–27]. Thus, to evince active inference in neural networks, it is necessary to demonstrate that they can plan to minimise future risk.”

“To address this issue, this work identifies a class of biologically plausible cost functions for two-layer recurrent neural networks, under an assumption that neural activity and plasticity minimise a common cost function (referred to as assumption 1). Then, we analytically and numerically demonstrate the implicit ability of neural networks to plan and minimise future risk, when viewed through the lens of active inference. ...”

In particular, one confusing fact is that the title suggests the paper main achievement is demonstrating that NNs perform active inference, but (unless I’m mistaken) this seems to be a corollary of the complete class theorem. I’m sorry to bother with this, but I fear that if I’m

confused with this, many readers will be as well.

[Response]

We aim to show that canonical neural networks perform active inference and its possible implementation by neuronal substrates such as delayed modulation of synaptic plasticity. They cannot be answered by simply applying the complete class theorem, because the complete class theorem guarantees only the existence of implicit Bayesian models. However, it does not specify the implementation. Thus, we believe that the title is apt, but would be happy to make this point explicit if you thought it would help.

2. The term “canonical neural network” has an important role in the paper, but I don’t see what is about them that deserves the term “canonical”. My belief is that most readers will believe “canonical NN” is a deep CNN or something of that sort. I’d suggest the authors to use another terminology, or to clarify their choice.

[Response]

Thank you for this comment. In the Introduction, we define our use of canonical neural networks as follows:

“Namely, we suppose a network of rate coding neurons with a sigmoid activation function, wherein the middle layer involves recurrent connections, and the output layer provides feedback responses to the environment (assumption 2). In this work, we will call such architectures canonical neural networks (**Table 1**).”

To further clarify the definition of terminology, we have added the following to **Table 1**:

Table 1. Glossary of expressions

Expression	Description
Canonical neural network	In this work, a canonical neural network is defined by differential equations of neural activity derived as a reduction of realistic neuron models through some approximations, which give a network of rate coding neurons with a sigmoid activation function. In particular, we consider networks comprising a middle layer that involves recurrent connections and the output layer that provides feedback responses to the environment.

3. I still believe the term “likelihood mapping” should be clarified. Additionally, the expression $p(o_\tau | s_\tau, A)$ suggests that A is a random variable, but it seems A is just a matrix of transition probabilities.

[Response]

Thank you for this comment. As we explained in the main text and **Table 1**, A is a random variable matrix that encodes the conditional probability. In this paper, we employ the following relationship to obtain the posterior expectation: $E_{Q(A)}[\ln P(o_\tau | s_\tau, A)] = \int \ln P(o_\tau | s_\tau, A) Q(A) dA = o_\tau \cdot \ln \mathbf{A} s_\tau$. Hence, we use the fact that o_τ is conditioned on the random (unknown) variable A as well as s_τ . Thus, this notation is apt to convey our intended meaning. Moreover, we usually use ‘transition probability’ to express a probability of state s_t being transformed to s_{t+1} . Thus, to avoid confusion, we instead call A as the likelihood mapping.

Also, note that $p(o_\tau | s_\tau, A)$ is a conditional probability which changes for each value of s_τ , hence I believe the notation $\text{cat}(A)$ suggest that it doesn’t depend on s_τ but only in A (if it would be a random variable).

[Response]

In this paper, $P(o_\tau | s_\tau, A) = \text{Cat}(A)$ means the conditional probability of o_τ given s_τ and A . The fact that A is a matrix indicates that $\text{Cat}(A)$ denotes the conditional probability. For example, if o_τ and s_τ take either 1 or 0, the conditional probability is fully expressed by a 2×2 matrix $A = \begin{pmatrix} A_{11} & A_{10} \\ A_{01} & A_{00} \end{pmatrix} = \begin{pmatrix} P(o_\tau = 1 | s_\tau = 1, A) & P(o_\tau = 1 | s_\tau = 0, A) \\ P(o_\tau = 0 | s_\tau = 1, A) & P(o_\tau = 0 | s_\tau = 0, A) \end{pmatrix}$. Thus, $\text{Cat}(A)$ is sufficient to express the conditional probability. However, we appreciate the reviewer’s concern. To address it, we have revised **Table 1** as follows:

Table 1. Glossary of expressions

Expression	Description
$\text{Cat}(A)$	Categorical distribution. In this expression, the probability that $o_\tau = i$ occurs given $s_\tau = j$ is A_{ij} ; that is, $P(o_\tau = i s_\tau = j, A) = A_{ij}$. For any pair of i and j , this is expressed as $P(o_\tau s_\tau, A) = \text{Cat}(A)$, where matrix A is the likelihood mapping that maps s_τ to o_τ . Although one may prefer to denote it as $\text{Cat}(A s_\tau)$ to emphasise that it changes depending on s_τ , in this paper, we use $\text{Cat}(A)$ following the notation in previous work. Note that $A_{kl}^{(i)}$ means that the probability that

	$o_\tau^{(i)} = k \text{ occurs given } s_\tau = \vec{l} = (l_1, \dots, l_N)^T \in \{0,1\}^{N_s}.$
--	--

One potential solution could be to use the notation $p(o_\tau | s_\tau; A)$, with the “;” separating random variables from parameters. Another possible solution would be to use A as a subscript of p . Beyond the specific solution, it has to be made clear how to differentiate between random variables and parameters.

[Response]

Thank you very much for pointing this out. Indeed, A (and B, C) are parameters of the POMDP model, which are random variables. Given that these are random variables, we have avoided the semicolon. When the agent performs variational inference, these parameters are inferred assuming a factorial distribution (see Eq.4). In order to avoid the confusion, we have added the following sentence when we introduced these parameters (see Eq. 2):

“where $\theta := \{A, B, C\}$ constitute the set of parameters and $P(s_1 | s_0, \delta_0, B) = P(s_1)$ and $P(\delta_1 | s_0, \gamma_t, C) = P(\delta_1)$ denote probabilities at $\tau = 1$. $P(\theta)$ and $P(\gamma_t)$ are the prior distributions of the parameters and the risk, which implies that θ and γ_t are treated as random variables in this work (**Table 1**).”

Additionally, from the table, what does it mean for a probability to become “realised”?

[Response]

We have revised **Table 1** by replacing “is realised” with “occurs” (see above).

Other points:

We have deleted a doubled reference.

We hope that these revisions are what you had in mind.

REVIEWERS' COMMENTS:

Reviewer #3 (Remarks to the Author):

The clarifications made greatly help the paper, I have no further suggestions. Thanks for the patience in addressing my comments.